# Past climate and continentality inferred from ice wedges at Batagay megaslump in the Northern Hemisphere's most continental region, Yana Highlands, interior Yakutia

Thomas Opel[1,2], Julian B. Murton[1], Sebastian Wetterich[2], Hanno Meyer[2], Kseniia Ashastina[3], Frank Günther[4,5,2], Hendrik Grotheer[6], Gesine Mollenhauer[6], Petr P. Danilov[7], Vasily Boeskorov[7], Grigoriy N. Savvinov[7], Lutz Schirrmeister[2]

[1]Permafrost Laboratory, Department of Geography, University of Sussex, Brighton, BN1 9RH, United Kingdom
[2]Alfred Wegener Institute, Helmholtz Centre for Polar and Marine Research, Telegrafenberg A45, 14473 Potsdam, Germany
[3]Senckenberg Research Institute and Natural History Museum, Research Station of Quaternary Palaeontology, 99423 Weimar, Germany
[4]Institute of Geosciences, University of Potsdam, Karl-Liebknecht-Str. 24-25, 14476 Potsdam, Germany
[5]Laboratory Geoecology of the North, Faculty of Geography, Lomonosov Moscow State University, Leninskie Gory 1, 119991 Moscow, Russia
[6]Alfred Wegener Institute, Helmholtz Centre for Polar and Marine Research, Am Handelshafen 12, 27570 Bremerhaven, Germany
[7]Science Research Institute of Applied Ecology of the North, North-East Federal University, 43 Lenin Avenue, Yakutsk 677007, Russia

*Correspondence to*: Thomas Opel (t.opel@sussex.ac.uk, thomas.opel@awi.de)

## Abstract

Ice wedges in the Yana Highlands of interior Yakutia—the most continental region of the Northern Hemisphere—were investigated to elucidate changes in winter climate and continentality since the Middle Pleistocene. The Batagay megaslump exposes ice wedges and composite wedges that were sampled from three cryostratigraphic units: the Lower Ice Complex of likely pre-Marine Isotope Stage (MIS) 6 age, the Upper Ice Complex (Yedoma) and the Upper Sand unit (both MIS 3 to 2). A terrace of the nearby Adycha River provides a Late Holocene (MIS 1) ice wedge that serves as a modern reference for interpretation. The stable-isotope composition of ice wedges in the MIS 3 Upper Ice Complex at Batagay is more depleted (mean $\delta^{18}O$ about −35‰) than that from 17 other ice-wedge study sites across coastal and central Yakutia. This observation points to lower winter temperatures and, therefore, higher continentality in the Yana Highlands during MIS 3. Likewise, more depleted isotope values are found in Holocene wedge ice (mean $\delta^{18}O$ about −29‰) compared to other sites in Yakutia. Ice-wedge isotopic signatures of the Lower Ice Complex (mean $\delta^{18}O$ about −33‰) and of the MIS 3–2 Upper Sand unit (mean $\delta^{18}O$ from about −33 to −30‰) are less distinctive regionally. The latter unit preserves traces of fast formation in rapidly accumulating sand sheets and of post-depositional isotopic fractionation.

# 1 Introduction

Interior Yakutia is currently the most continental region of the Northern Hemisphere. At Verkhoyansk, the pole of cold of the Northern Hemisphere, mean monthly air temperature ranges exceed 60°C and absolute temperatures ranges can reach 100°C (Lydolph, 1985). Mean annual precipitation is only 155 mm (Lydolph, 1985). Scarce stable-isotope data indicate that modern winter precipitation in this region has the most depleted isotopic composition in Siberia (Kurita et al., 2004; Kurita et al., 2005) according to the "continental effect" in stable isotope fractionation, i.e. the farthest distance from the source region (Dansgaard, 1964). However, warm summers enrich stable isotopes in summer precipitation at interior sites compared to coastal sites in the North, which also reflects the strong continentality of interior Yakutia. During Late Pleistocene cold stages, when lower sea level exposed larger areas of continental shelves, interior Yakutia probably experienced even stronger continentality, with further decreased annual precipitation and increased annual temperature amplitudes. Such increased amplitudes must have resulted from either lower winter temperatures or higher summer temperatures or a combination of both.

While biological proxies such as pollen or plant macrofossil remains can be used to reconstruct summer climate, winter conditions in interior Yakutia may be inferred only from ice wedges, which can be found in Middle and Late Pleistocene as well as Holocene permafrost deposits. Ice wedges provide winter temperature information due to their specific seasonality, with frost cracking in winter and crack infilling by snowmelt in spring (Opel et al., 2018). They integrate the stable-isotope composition of cold-season precipitation (Meyer et al., 2015), which in high northern latitudes is mainly a function of temperature (Dansgaard, 1964). However, a knowledge gap exists about the winter palaeoclimate of interior Yakutia, because ice-wedge studies in the Yana Highlands—between the Verkhoyansky and Chersky ranges—began only recently (Vasil'chuk et al., 2017). In contrast, other sites in east Siberia (west Beringia) have been extensively studied over recent decades, including the coastal lowlands to the north (e.g. Meyer et al., 2002a; Meyer et al., 2002b; Opel et al., 2017b), the Kolyma region to the east (e.g. Vasil'chuk et al., 2001; Vasil'chuk, 2013; Vasil'chuk and Vasil'chuk, 2014) as well as central Yakutia to the south (e.g. Popp et al., 2006) (Figure 1). Ice-wedge studies have been carried out also in east Beringia, e.g. in Alaska, Yukon, and Northwest Territories (Kotler and Burn, 2000; Meyer et al., 2010b; Fritz et al., 2012; Lachniet et al., 2012; Porter et al., 2016). Here we report cryostratigraphic field observations of ice wedges and composite wedges made during a reconnaissance expedition from 27 July to 5 August 2017 together with stable-isotope data from the Batagay megaslump and Adycha River floodplain (Figure 1). New radiocarbon ages from organic remains in ice wedges and host sediments help constrain the late Quaternary chronology. The observations and ages elucidate the history of wedge development, past winter temperatures and continentality since the Middle Pleistocene.

# 2 Study site characteristics, stratigraphy and chronology

Neither study site has been glaciated during at least the last 50,000 years, and the Batagay megaslump provides access to permafrost formations since the Middle Pleistocene (Ashastina et al., 2017; Murton et al., 2017). The present mean annual air temperature at Batagay is –15.4°C (mean of the coldest month: –47.7°C, mean of the warmest month: 15.4°C) and the mean

annual precipitation is 170 to 200 mm (Ivanova, 2003). Prevailing wind directions are southwest in winter (January) and north or northeast in summer (July) (Murton et al., 2017). The mean annual temperature of the permafrost is −8.0 to −5.5°C and the active-layer depth between 0.2–0.4 m beneath forest and moss covers and 0.4–1.2 m beneath open sites (Ivanova, 2003).

The **Batagay megaslump** (67.58°N, 134.77°E), close to the city of Batagay on the east bank of the Yana River, is the world's
largest known thaw slump (Figure 2) (Ashastina et al., 2017; Murton et al., 2017). It is located on a hillslope in the taiga landscape (328 m above sea level for the highest part of the headwall) and exposes a sequence of permafrost deposits at least 60 m thick, with underlying bedrock cropping out in places (Kunitsky et al., 2013). The slump has formed during recent decades (Savvinov et al., 2018), with increasing headwall retreat rates of up to 30 m per year. In 2016 it reached a width of 840 m and a total area of >70 ha (Günther et al., 2016).

The slump exposes Pleistocene and Holocene permafrost formations ranging in age from MIS 6 (or older) to MIS 1 (Ashastina et al., 2017; Murton et al., 2017; Ashastina et al., 2018). Above slate bedrock and a basal diamicton, four major cryostratigraphic units contain ice wedges and/or composite wedges (Figure 3, Table 1).

The lowest unit is a pebbly dark sand 3–7 m thick, with ice wedges at least 2–3 m high and 1 m wide truncated by a thaw unconformity. This **Lower Ice Complex** has neither been dated nor sampled previously for wedge ice.

The **Lower Sand** unit above reaches about 20 m in thickness and is composed of yellowish pore-ice cemented (fine) sand with grey horizontal bands. It contains tall, narrow syngenetic ice wedges up to 0.5 m wide. The middle part of the unit was dated to 142.8±25.3 and >123.2 kyr by optically-stimulated luminescence (OSL) and to 210.0±23.0 kyr by infrared-stimulated luminescence (IRSL) (Ashastina et al., 2017), pointing to its deposition during MIS 6. This age is supported by palaeoecological data (Ashastina et al., 2018). An organic-rich unit of silty sand with abundant wood and plant remains (in
lenses up to 3 m thick) overlies a distinct erosional surface near the top of the Lower Sand unit. A wood fragment yielded a conventional radiocarbon age of 49.32±3.15 [14]C kyr BP, outside of the range of calibration (Murton et al., 2017), though palaeoecological analysis points to a Last Interglacial age for the organic material (Ashastina et al., 2018).

The overlying **Upper Ice Complex** is 20–25 m thick and dominated by huge syngenetic ice wedges up to few metres wide and at least several metres high within silty and sandy deposits. Finite radiocarbon ages from 49.0±2.0 to 12.66±0.05 [14]C kyr
BP (Ashastina et al., 2017) indicate sedimentation gaps or erosive events and reveal a MIS 3 to MIS 2 age for the upper part of the Upper Ice Complex. This unit has numerous chronostratigraphic analogies in the coastal lowlands of northern Yakutia, i.e. the Yedoma Ice Complex (Schirrmeister et al., 2011b; Murton et al., 2015). The Upper Ice Complex forms the highest Pleistocene stratigraphic unit in the upper central part (upslope) of the slump headwall, whereas downslope towards the slump mouth it is overlain and partially grades into the **Upper Sand unit** (Figure 3).

The Upper Sand is up to about 20 m thick and consists of pore-ice cemented brown to grey sand with narrow (≤ 0.5 m wide) syngenetic ice wedges and composite wedges. Radiocarbon ages between 36.30±0.70 [14]C kyr BP and 26.18±0.22 [14]C kyr BP (Ashastina et al., 2017; Murton et al., 2017) indicate deposition during MIS 3 and MIS 2, similar to the Upper Ice Complex, but in a more complex spatio-temporal pattern, likely due to varying palaeotopography, sediment supply, and accumulation rates.

A near-surface layer 1–1.5 m thick of brown sand and modern soil covers the sequence. This layer was dated to 295±30 [14]C yr BP (Ashastina et al., 2017). Distinct (thermo-)erosional contacts can be found between the Lower Ice Complex and the Lower Sand, the Lower Sand and the Upper Ice Complex and below the near-surface layer.

According to Murton et al. (2017) exposed floodplains of proximal rivers such as the Batagay and Yana, within 2 and 10 km, respectively, of the slump are the assumed major source of the sediments exposed in the headwall, which implies upslope directed transport by wind. Periglacial and nival processes on nearby hillslopes may also have contributed to sediment supply (Ashastina et al., 2017).

For a late Holocene reference, we studied an ice wedge at the actively eroding Holocene to recent bank of a small cut-off channel of the **Adycha River** (67.66°N, 135.69°E, 138 m above sea level), about 40 km east of the Batagay megaslump (Figure 1). The Adycha site is located about 3 km upstream of the 60–65 m high and 1.2 km long river-bluff stratigraphic section at Ulakhan Sullar (Kaplina et al., 1983; Sher et al., 2011; Germonpre et al., 2017). Organic-rich silty sands of alluvial origin are exposed in the Adycha riverbank section.

## 3 Material and Methods

### 3.1 Fieldwork

Cryostratigraphic observations of ice wedges, composite wedges, unit contacts and sediments at the Batagay megaslump and a terrace of the Adycha River provided a framework for sample selection. A total of six ice wedges and composite wedges from the Lower Ice Complex, the Upper Ice Complex and the Upper Sand were described in detail and sampled at two sections of the slump (Figures 2, 3). Section 1 was studied in a gully in the icy badlands of the slump (Figure 4), and Section 2 was studied in a headwall slope segment (Figure 5). Composite wedges of the Lower Sand unit were inaccessible due to dangerous outcrop conditions, and no wedges were observed in the near-surface layer. At the Adycha River, we sampled a single ice wedge.

Ice samples about 4 cm wide, mostly in horizontal profiles, were obtained by chain saw. Details about the studied ice and composite wedges are given in Table S1. The ice samples were melted on site in freshly opened standard Whirl-Pak sample bags, and the meltwater filled up 30 ml PE bottles that were then tightly closed and stored cool until stable-isotope analysis.

In addition, water from small streams draining the slump was sampled in several parts of the slump floor and the main outflow stream (n=7) as well as from summer precipitation (n=4). Supernatant water from selected Batagay megaslump host sediment samples (n=9) was taken for stable-isotope analysis of intrasedimental pore and segregated ice.

### 3.2 Stable-isotope analysis

The stable oxygen ($\delta^{18}O$) and hydrogen ($\delta D$) isotope ratios of all samples were determined at the Stable Isotope Laboratory of the Alfred Wegener Institute Helmholtz Centre for Polar and Marine Research in Potsdam, using a Finnigan MAT Delta S mass spectrometer with an analytical precision of better than ±0.1‰ for $\delta^{18}O$ and ±0.8‰ for $\delta D$ (Meyer et al., 2000). The

isotopic composition is expressed in $\delta$ per mille values (‰) relative to the V-SMOW standard. The deuterium excess $d$ was calculated as $d = \delta D - 8 \times \delta^{18}O$ (Dansgaard, 1964). For three of seven studied wedges, samples were excluded from further analysis (five in total, Table S2) on the basis that they were lateral samples with high sediment content and/or had isotopic values distinctly different from the wedge centres and similar to host sediments, probably due to exchange processes between

wedge ice and host sediments (Meyer et al., 2002a; Meyer et al., 2010a).

## 3.3 Radiocarbon dating

We picked organic material from our ice-wedge and host sediment samples for radiocarbon dating at the MICADAS [14]C dating facility of the Alfred Wegener Institute Helmholtz Centre for Polar and Marine Research in Bremerhaven. The organic material was subjected to chemical leaching and cleaning following the ABA (acid–base–acid) procedure. Material was first submerged

in 1 M HCl (for 30 min at 60 °C) to remove carbonate contamination. Following rinsing with Milli-Q water, samples were submerged in 1 M NaOH (for 30 min at 60 °C) to leach out humic acids. The base leaching was repeated for a minimum of 7 times or until no coloring of the solution was evident. Afterwards the material was subjected to a final acid treatment (1 M HCl; 30 min at 60 °C), rinsed to neutral with Milli-Q and dried for a minimum of 12 h at 60 °C. Milli-Q used for cleaning and acid/base solutions was pre-cleaned by liquid–liquid extraction with dichloromethane to remove remaining organic

contamination.

Samples were packed in tin capsules and combusted individually using an Elementar vario ISOTOPE EA (Elemental Analyzer). If samples were smaller than approximately 200 µgC, the $CO_2$ produced by combustion with the EA was directly injected into the hybrid ion source of the Ionplus MICADAS using the gas interface system GIS (Fahrni et al., 2013). For larger samples, the $CO_2$ produced was graphitized using the Ionplus AGE3 system (Automated Graphitization System;

(Wacker et al., 2010c). The radiocarbon content of the samples was determined alongside size-matched reference standards (oxalic acid; NIST 4990c) and blanks (phthalic anhydride; Sigma-Aldrich 320065) using the Ionplus MICADAS dating system (Synal et al., 2007; Wacker et al., 2010a), and blank correction and standard normalization were performed using the BATS software (Wacker et al., 2010b). Results are reported as $F^{14}C$ (Reimer et al., 2004) and conventional radiocarbon ($^{14}C$) ages in years BP. Several analyses yielded lower $^{14}C$ intensities than the blanks included in the respective sequence. In these cases,

results are reported as $F^{14}C_{sample} < F^{14}C_{blanks}$. Conventional $^{14}C$ ages were calibrated using Oxcal 4.3 (Bronk Ramsey, 2009) based on the IntCAL13 dataset (Reimer et al., 2013).

## 4 Results

### 4.1 Batagay megaslump

#### 4.1.1 Field observations of ice wedges

The present study confirms cryostratigraphic observations about ice wedges and composite wedges from earlier studies (Ashastina et al., 2017; Murton et al., 2017) and provides new observations about ice-wedge relationships across contacts between the cryostratigraphic units, as outlined below.

In the Lower Ice Complex, ice wedges are truncated by a thaw unconformity, though the toes of some narrow syngenetic composite wedges in the overlying Lower Sand unit extend down across the contact (Figure S1). A single example of clear ice sharply overlying the shoulder of an ice wedge in the Lower Ice Complex (B17-IW1, Figure 4) contained brown plant remains and round, few-mm diameter organic bodies identified as hare droppings. This ice lacked the foliation characteristic of wedge ice and is interpreted as pool ice. The basal contact of the Lower Ice Complex was not observed.

In the Lower and Upper Sand units, narrow syngenetic wedges vary in apparent width from a few centimetres to about 0.5 m. Width commonly varies irregularly with height along individual wedges, sometimes gradually, sometimes abruptly. Wedge height varies from a few metres to at least 12 m. Wedges tend to be oriented approximately at right angle to colour bands in the sand, such that wedges in the Upper Sand unit—whose bands dip downslope, parallel to the ground surface—are characteristically subvertical, with their tops inclined downslope (Figure S2). The wedge infills grade between end members of ice-wedge ice and icy sand wedges, with slightly sandy ice-wedge ice (i.e. composite wedges) the most common type.

Near the base of the Upper Ice Complex, syngenetic ice wedges, a few metres wide tend to narrow downwards in the lower several metres of the unit and terminate with irregular or flattish or U- to V-shaped toes over a vertical distance of about 1–3 m (Figure S2). At the northeast part of the slump headwall, however, the base of the Upper Ice Complex is a sharp contact that cuts down into and truncates bands in the underlying Lower Sand unit over a vertical distance of up to at least several metres (Figure 3). As a result, the Upper Ice Complex thickens substantially downslope.

The top of the Upper Ice Complex displays a variety of wedge relationships with the overlying Upper Sand unit (Figure S3). Some wide syngenetic wedges terminate along planar to gently undulating contacts that are horizontal to gently dipping and have apparent widths of about 1–3 m; some taper upward into narrow wedges characteristic of the Upper Sand unit; some taper irregularly upward, marked by shoulders up to about 0.5 m wide; and some end upwards with a narrow offshoot. These changes occur over a vertical distance of about 1–3 m.

#### 4.1.2 Lower Ice Complex (pre-MIS 6)

A small truncated ice wedge B17-IW1 (0.5 m wide, 1.1 m high) from the Lower Ice Complex was sampled in section 1 (Figure 4) near the slump bottom, about 50 m below the ground surface and about 5 m below the altitude level in the Lower Sand unit dated by luminescence to 142.8±25.3 and >123.2 kyr (OSL) and to 210.0±23.0 kyr (IRSL) (Ashastina et al., 2017). It shows a

mean stable-isotope composition of –33.1‰ for $\delta^{18}O$, –256.7‰ for $\delta D$, and 8.0‰ for $d$ (n=6), with very little variability (Table 2, Figure 6). The regression line in a $\delta^{18}O$–$\delta D$ bi-plot is $\delta D = 12.78\ \delta^{18}O + 166.23$ ($R^2 = 0.80$).

The isotopic composition of the intrasedimental ice (n=2) shows similar mean $\delta$ values (–32.5‰ for $\delta^{18}O$, –247.1‰ for $\delta D$), but higher $d$ values (12.8‰). The co-isotopic regression line is $\delta D = 4.09\ \delta^{18}O - 114.27$.

Radiocarbon dating of twigs, Cyperaceae stems and roots from a sediment sample 0.5 m above the truncated ice wedge yielded nonfinite ages of >53,400 and >37,500 [14]C yr BP (Table 3). A hare dropping from the ice wedge was dated to 15,792±358 [14]C yr BP, which can be only explained by contamination with relocated material.

### 4.1.3 Lower Sand unit (MIS 6)

The isotopic composition of intrasedimental ice (n=2) from the Lower Sand unit shows very little variability (mean –32.3‰
for $\delta^{18}O$, –241.6‰ for $\delta D$, 16.6‰ for $d$ (Table 2, Figure 6). The co-isotopic regression line is $\delta D = 1.33\ \delta^{18}O – 198.54$.

### 4.1.4 Upper Ice Complex (MIS 3–2)

Two large syngenetic ice wedges from the overlying ice-rich Upper Ice Complex in section 2 (Figure 5) in the southern part of the slump show more depleted isotope values as compared to the Lower Ice Complex (Table 2, Figure 6). Ice wedge B17-IW6 (about 0.5 m wide, sampled about 26 m below surface, bs) shows a mean isotopic composition of –35.1‰ for $\delta^{18}O$, –
269.4‰ for $\delta D$, and 11.0‰ for $d$ (n=4). The co-isotopic regression is $\delta D = 12.25\ \delta^{18}O + 159.79$ ($R^2 = 0.99$). The mean stable-isotope values of ice wedge B17-IW5 (about 1.6 m wide and sampled about 20 m bs) are –34.9‰ for $\delta^{18}O$, –271.0‰ for $\delta D$, and 8.1‰ for $d$ (n=12). The regression line in a $\delta^{18}O$–$\delta D$ bi-plot is $\delta D = 7.41\ \delta^{18}O – 12.36$ ($R^2 = 0.99$). The intrasedimental ice data (n=2) show distinctly enriched mean values (–26.2‰ for $\delta^{18}O$, –195.0‰ for $\delta D$, 14.5‰ for $d$).

Unidentified plant (bract fragments and roots) and insect (complete pieces and fragments of elytron) remains from ice wedge
B17-IW6 as well as Cyperaceae remains and roots from host sediment at the lowest studied level in section 2, about 26 m bs, both revealed nonfinite ages (>37,500 [14]C yr BP; Table 3). At the level of about 20 m bs rootlets from the host sediment yielded ages of >37,500 and 47,550±677 [14]C yr BP, whereas unidentified plant remains (twigs, roots and florets) from within ice wedge B17-IW5 were dated to 24,858±536 [14]C yr BP.

### 4.1.5 Upper Sand unit (MIS 3-2)

Two narrow sand–ice wedges (composite wedges) and one ice wedge were sampled from the rather ice-poor Upper Sand unit in section 2 (Figure 5) at the southern part of the slump. Composite wedge B17-IW2 (0.25 m wide) was sampled 2.6 and 2.8 m bs, and composite wedge B17-IW3 (0.2 m wide) was sampled 2.0 m bs. The stable-isotope composition of the composite wedges from the Upper Sand unit differs from those of the Lower Ice Complex and the Upper Ice Complex (Table 2, Figure 6). The mean values are –32.8‰ and –31.9‰ for $\delta^{18}O$, –247.5‰ and –240.5‰ for $\delta D$, and 15.1‰ and 14.7‰ for $d$ (n=4 and
n=2), respectively, and plot well above the Global Meteoric Water Line (GMWL, $\delta D = 8\ \delta^{18}O + 10$) (Craig, 1961). The corresponding regressions are $\delta D = 12.51\ \delta^{18}O + 163.29$ ($R^2 = 0.92$) and $\delta D = 14.47\ \delta^{18}O + 220.91$, respectively.

Ice wedge B17-IW4 (about 0.4 m wide and containing a few thin sand veins) was sampled in the Upper Sand 2.0 m bs and exhibits a more enriched stable-isotope composition (mean values –29.9‰ for $\delta^{18}O$, –227.8‰ for $\delta D$, and 11.6‰ for $d$ (n=6). The co-isotopic regression is $\delta D = 15.03\ \delta^{18}O + 222.01$ ($R^2 = 0.91$) (Table 2, Figure 6).

The intrasedimental ice (n=3) shows similar mean $\delta$ values (–30.6‰ for $\delta^{18}O$, –220.0‰ for $\delta D$) but higher $d$ (24.4‰) values compared to the composite wedges and the ice wedges of the Upper Sand. The regression line in a $\delta^{18}O$–$\delta D$ bi-plot is $\delta D = 10.61\ \delta^{18}O + 104.19$ ($R^2 = 0.99$).

Two radiocarbon ages (analysed as gas) of small samples of *Empetrum nigrum* (crowberry) leaf, twigs, and Cyperaceae remains from host sediment about 11 m bs are nonfinite, whereas a graphite target yielded an age of 38,348±236 [14]C yr BP.

### 4.1.6 Slump outflow

The samples from the outflow stream draining the slump show average stable-isotope values of –31.7‰ for $\delta^{18}O$, –238.6‰ for $\delta D$, and 15.2‰ for $d$. . They plot on a co-isotopic regression line of $\delta D = 8.84\ \delta^{18}O + 41.83$ ($R^2 = 0.97$, n=7), well above the GMWL (Table 2, Figure 6) .

### 4.1.7 Summer precipitation

The samples of summer precipitation at the Batagay megaslump show much more enriched stable-isotope compositions than the wedges, as expected (Table 2). Mean values are –14.3‰ for $\delta^{18}O$, –112.0‰ for $\delta D$, and 2.6‰ for $d$ (n=4). The samples plot on a co-isotopic regression line of $\delta D = 7.93\ \delta^{18}O + 1.65$ ($R^2 = 1$).

### 4.2 Adycha River

Ice wedge A17-IW3 was about 0.3 m wide and sampled about 1.4 m above river level and about 2 m below the ground surface. Its mean stable isotope values are –29.0‰ for $\delta^{18}O$, –226.3‰ for $\delta D$, and 5.5‰ for $d$ (n=10, Table 2, Figure 6). The regression line in a $\delta^{18}O$–$\delta D$ bi-plot is $\delta D = 7.01\ \delta^{18}O - 23.25$ ($R^2 = 0.99$).

Rootlets and larch needles from the host sediment were dated to 213±109 [14]C yr BP and confirm the Late Holocene age of the alluvial sediments and the ice wedge.

### 5 Discussion

### 5.1 Chronostratigraphy of the Batagay megaslump

To date, the exceptionally long record of permafrost history under highly continental climate conditions as exposed in the Batagay megaslump is still not entirely established in terms of its geochronology. Previous studies by Ashastina et al. (2017) and Murton et al. (2017) provide the general differentiation into four main units with ice and/or composite wedges overlain by Late Holocene cover deposits, which are the Lower Ice Complex (pre-MIS 6), the Lower Sand unit (MIS 6), the Upper Ice

Complex (MIS 3–2) and the Upper Sand unit (MIS 3–2). Thaw unconformities as observed in sampled sections of this study (Figure 4, Figure 5), but also in previous studies (Ashastina et al., 2017; Murton et al., 2017) question the chronological continuity of the archive. Such (thermo-)erosional features are often observed in late Quaternary permafrost chronologies (e.g. Wetterich et al., 2009) and most likely caused by intense permafrost degradation during warm stages such as the Last

Interglacial with widespread thermokarst (Reyes et al., 2010; e.g. Kienast et al., 2011) or exceptionally warm interstadials, but might also relate to palaeotopography and respective accumulation and erosion areas.

The Batagay megaslump, on account of its huge dimensions and rapid erosion, holds potential to identify additional records and units in future studies. With the present study and additional radiocarbon dating, further confidence of the existing stratigraphy is reached, although the wedge-ice derived age information requires careful interpretation. Due to the formation

mechanism and downward directed growth of wedge ice, organic remains from inside the ice are commonly younger than those of the host deposits at the same altitude (i.e. sampling depth). The risk of dating redeposited organic material, that entered into or later froze onto the surface of ice recently exposed in a slump, is common, in particular in erosional features such as slump-floor gullies. It can be seen in the truncated **Lower Ice Complex** (Figure 4) where an age of about 16 [14]C kyr BP was obtained from the ice wedge, while radiocarbon dates of the host deposit are infinite and luminescence ages of the overlying

Lower Sand unit reach 142.8±25.3 kyr and >123.2 kyr by OSL as well as 210.0±23.0 kyr by IRSL (Ashastina et al., 2017). We therefore excluded this anomalously young radiocarbon age from further interpretation.

A distinct unit of wood and plant remains up to 3 m thick and traceable along the exposure is situated near the top of the Lower Sand unit and below the Upper Ice Complex. Radiocarbon dating of this layer revealed an infinite radiocarbon age of >44 [14]C kyr BP (Ashastina et al., 2017) as well as an out-of-calibration age of 49.32±3.15 [14]C kyr BP (Murton et al., 2017). The

palaeobotanic proxy data indicate a warm climate. The plant macro-fossil record is characterized by considerable species richness (34 species), mainly representing northern Taiga forests, including larch and birch trees, shrub alder and raspberry and further representatives of the taiga's understorey. Hence the unit is aligned to the Last Interglacial (Ashastina et al., 2018), which caused particular thaw of the underlying Lower Sand unit and formed a disconformity.

The expected difference between ice-hosted and sediment-hosted ages of the same sample depth is clearly seen in the **Upper**

**Ice Complex**, where organic remains from the ice wedge are dated to about 25 [14]C kyr BP and the host deposits yield nonfinite to about 48 [14]C kyr BP ages (Figure 5). However, even though the age offset is very large and a contamination of the dated sample in the field cannot be fully excluded, such information may be reliable and lies within the previously dated overall accumulation period of the Upper Ice Complex between about 49 and 13 [14]C kyr BP (Ashastina et al., 2017; Murton et al., 2017).

The new radiocarbon ages from the **Upper Sand unit** largely confirm the assumed accumulation period of this unit from about 36 to 26 [14]C kyr BP (Ashastina et al., 2017; Murton et al., 2017), but predate it by about 2000 years, at least with a date of about 38 [14]C kyr BP and nonfinite ages of >37.5 [14]C kyr BP from the host deposits. We note that the Upper Sand unit has been studied at two different sites of the slump in different altitude levels (north-northeast of the central headwall (Ashastina et al., 2017; Murton et al., 2017) and together with the Upper Ice Complex more downslope in the southern part of the slump (Figure

2), which complicates the correlation of dating results from different sampling sites. Furthermore, close to our section 2, the Upper Sand could not clearly be differentiated from the Upper Ice Complex during fieldwork in 2014 (Ashastina et al., 2017). Hence, the more or less simultaneous accumulation of both the Upper Ice Complex and the Upper Sand unit during MIS 3–2 but their differing spatial distribution in the slump highlights the importance of palaeo-topography in a hillslope setting and

varying sediment sources and velocities of deposition and require future sampling and analysis at higher vertical resolution. Interestingly, no indications for an Upper Sand unit have been found in the top of the modern headwall on the upslope part of the slump. Additionally, there are no significant Holocene deposits exposed at all in the upper part of the slump. However, carcasses of *Equus lenensis* and juvenile *Bison priscus* (dated to 4.45±0.35 and 8.22 +0.45/-0.40 kyr [14]C BP, respectively) were found at the exit of the slump (Murton et al., 2017). These observations may be related to (a) the palaeo-topographic

situation and transport and deposition regimes of windblown material during MIS 3–1, likely decreasing upslope, (b) widespread permafrost degradation since the Late Glacial–Early Holocene transition, and/or (c) changed hydrological and vegetation patterns in the Holocene preventing transport and/or deposition of windblown sediment.

The **Lower Ice Complex** is not dated so far. Given the dating results of the overlying Lower Sand unit it might correspond to the Yukagir Ice Complex from Bol'shoy Lyakhovsky Island, which has been dated by radioisotope disequilibria ([230]Th/U) in

peat to 178±14 kyr, 221±27 kyr (Wetterich et al., 2019) and 201±3 kyr (Schirrmeister et al., 2002). Hence this unit most likely has survived both the last and current interglacials. Dating such an old Ice Complex is challenging as the dating approaches are limited and partly still in development. Independently dated tephra layers have been proven useful to date old permafrost in east Beringia, i.e. in Alaska (Schirrmeister et al., 2016) and in Yukon (Froese et al., 2008). However, no tephra layers have been identified yet in the study region due to the long distance from active volcanos. But with its clearly defined stratigraphy

and the access to very old permafrost and ice wedges, the Batagay megaslump may be a suitable site to evaluate and further develop the ice-wedge dating approaches using [36]Cl/Cl[-] (Blinov et al., 2009) as well as uranium isotope methods (Ewing et al., 2015). To better constrain the chronostratigraphy of the exposed units a systematic dating approach is needed, which should additionally include independent dating of host sediments, e.g. by luminescence.

**5.2 Cryostratigraphy at the Batagay megaslump – implications for ice-wedge growth regime**

The shape and composition of the wedges in the Batagay megaslump correspond to the assumed genesis of the respective units and may also provide palaeoclimate information (Table 4). Both the Lower and Upper Ice Complex units are characterized by ice wedges up to several m wide, whereas both the Lower and Upper Sand units contain tall narrow wedges. The former indicates more stable surface conditions with lower rates of sand aggradation and sufficient melt water supply during the formation of both Ice Complexes that allowed more horizontal growth of ice wedges. The tall narrow wedges of both sand

units, in contrast, point to rapid upslope-directed aeolian deposition of sand and, therefore, a predominant vertical ice-wedge growth in persistent polygonal patterns. The downslope inclination of narrow syngenetic wedges in the Upper Sand unit indicates upward growth subvertically at right angle to the aggrading depositional (hillslope) surface. The tall and narrow wedges in the sand units are similar in size and shape to chimney-like sand wedges in thick aeolian sand sheets in the

Tuktoyaktuk Coastlands of western Arctic Canada, attributed to rapid aggradation of aeolian sand and rapid wedge growth (Murton and Bateman, 2007). The composite wedges in the Lower and Upper Sand units imply dry conditions with only little melt water supply. However, as most ice wedges exhibit a significant sediment content, a steady supply of windblown sediment and a rather thin snow cover during ice-wedge formation is likely.

At the top of the Upper Ice Complex, ice wedges tend to narrow and partly transform into narrow composite wedges of the Upper Sand unit (Figure S3). This indicates a rather gradual change of the deposition regime towards higher sedimentation rates and possibly lower melt water supply due to drier conditions. Interestingly, the polygonal pattern has not been affected by this change, as indicated by the consistent frost-cracking positions. The upward transition of wedges from the Upper Ice Complex to Upper Sand unit, however, was interrupted episodically by thaw, producing a number of thaw unconformities at different depths. In contrast, the erosional event truncating the Lower Ice Complex seems to have changed the polygonal pattern in which the Lower Sand unit has been deposited.

A major erosional surface attributed to gullying by water flowing down a palaeo-hillslope is inferred from the large concave-up lower contact of the Upper Ice Complex in the northeast part of the headwall (Figure 3). Numerous gullies on the present hillslope near the megaslump are indicated on satellite images between 1968 and 2010 (Kunitsky et al., 2013), which suggests that gullies are characteristic landforms of such terrain under present environmental conditions. Water is supplied to such gullies by snowmelt in spring and rainfall and melt of ground ice in summer. Erosion of the underlying Lower Sand unit provided substantial accommodation space for development of an unusually thick Upper Ice Complex with wide ice wedges above it. Stratigraphically, this erosional surface is at a similar level as the upper woody debris layer that is thought to be of last interglacial age (Ashastina et al., 2017; Ashastina et al., 2018), which in turn suggests that the erosion also probably took place during warm (i.e. interglacial) conditions.

**5.3 Yana Highlands' ice-wedge stable isotopes – regional palaeoclimate implications**

Unfortunately, no comprehensive modern precipitation stable-isotope data are available for the Yana Highlands. The nearest sites with available data are Zhigansk, about 500 km to the west (Kurita, 2011); Tiksi, about 500 km to the northwest (Kloss, 2008); and Yakutsk, about 650 km to the south (Kloss, 2008; Papina et al., 2017). These sites show similar isotopic characteristics for the cold season (October to March). Mean $\delta^{18}O$ values decrease along the north–south transect from about –29‰ in Tiksi to –31‰ in Yakutsk, and $d$ varies strongly without a clear geographical pattern and shows values between about 8 and 15‰. In general, the mean winter $\delta$-values plot around the GMWL. However, these sites are characterized by less cold winters compared to the Yana Highlands. Furthermore, they are located east and south, respectively, of the Verkhoyansky Range (Figure 1), which constitutes an orographic barrier for atmospheric moisture transport by the westerlies and thereby contributes to the extreme cold and dry winters of the Yana Highlands. Hence, more depleted $\delta^{18}O$ values can be expected for winter precipitation.

The Yana Highlands' ice-wedge stable-isotope data form two major clusters (Figure 6, Table 2). The regression of the first cluster (cluster 1) plots below but mainly parallel to the GMWL ($\delta D = 7.45 \delta^{18}O – 10.46$, $R^2=1$) and comprises ice wedges

B17-IW1 (Lower Ice Complex), B17-IW5, B17-IW6 (both Upper Ice Complex) and A17-IW3 (recent Adycha riverbank). The regression of the second cluster (cluster 2) plots clearly above the GMWL ($\delta D = 6.88\ \delta^{18}O - 21.72$, $R^2=0.97$) and comprises all data from the Upper Sand unit (composite wedges B17-IW2, B17-IW3, ice wedge B17-IW4). Furthermore, all except one of the intrasedimental-ice samples as well as all slump outflow samples—interpreted as integrated signal of the entire exposed

Batagay sequence—plot well above the GMWL. In summary, the higher the sediment content of our samples, the higher are the $d$ values and, hence, the offset above the GMWL. The composite and ice wedges from the Upper Sand unit, likely formed in a generally dry environment (section 5.2), exhibit high $d$ values (cluster 2). In contrast, the ice wedges from cluster 1, likely formed under moister conditions with more abundant melt water supply, show lower $d$ values. This suggests that the divergence from cluster 1 (or the GMWL), i.e. higher $d$, might be an indicator of drier conditions.

Generally, the high $d$ values indicate a higher kinetic fractionation at the moisture source, e.g. due to initial evaporation under low relative humidity and/or high sea-surface temperature conditions (Pfahl and Sodemann, 2014). As we are dealing with cold-stage MIS 3–2 ground ice, a changed moisture source region might be more likely than generally higher sea-surface temperatures compared to the Late Holocene as reflected by the ice wedge A17-IW3. As the cold season in east Siberia is characterized by higher $d$ values than the warm season, one might conclude that cold-season precipitation does not only feed

the ice and composite wedges but also contributes significantly to the formation of intrasedimental ice. However, secondary fractionation during snow cover evolution under extremely continental conditions with a generally lower snow cover, and snow melt containing only a fraction of the initial snow might also affect $d$ preserved in wedge ice (Opel et al., 2018; Grinter et al., 2019). For intrasedimental ice, too, multiple freeze–thaw cycles in the active layer have to be taken into account.

The regional palaeoclimatic implications for the Yana Highlands drawn from our ice-wedge $\delta^{18}O$ data are summarised in Table

4 together with palaeoecological results from Ashastina et al. (2018). The lowest winter temperatures of our dataset are indicated by $\delta^{18}O$ values around –35‰ for ice wedges from the Upper Ice Complex, which formed during the MIS 3 interstadial (Kargin interstadial in Siberian stratigraphy). Similar values including $d$ have been reported by Vasil'chuk et al. (2017) for three undated ice wedges from the Upper Ice Complex sampled close to the headwall of the Batagay megaslump (Figure 3). Slightly higher temperatures during the formation of the Lower Ice Complex before MIS 6 are indicated by $\delta^{18}O$ values around

–33‰. However, given the small size of the corresponding ice wedge, the old age and rather similar isotope values of intrasedimental ice in the host deposits some postdepositional alteration of the originally recorded signal (Meyer et al., 2002a; Meyer et al., 2010a) cannot be excluded. Hence, these values should be treated with caution. Comparable and slightly higher winter temperatures can be inferred for composite wedges ($\delta^{18}O$ values around –33‰ and –32‰) and an ice wedge ($\delta^{18}O$ values about –30‰) from the Upper Sand unit, which is attributed to the transition from MIS 3 to 2. Again, similar values

have been reported by Vasil'chuk et al. (2017) for an undated narrow ice wedge from the Upper Sand unit sampled in another part of the Batagay megaslump (Figure 3). However, the palaeoclimatic significance of the rather small composite wedges with few samples has to be treated with caution. Ice wedge B17-IW4 might even be an epigenetic ice wedge of Holocene age, given the quite similar $\delta^{18}O$ values (about –29‰) of the definitively Holocene ice wedge from the Adycha river bank (A17-IW3), which indicate higher winter temperatures in the last centuries to millennia. These values are close to those of modern

winter precipitation at Tiksi, Zhigansk and Yakutsk. As modern ice wedges are enriched by several per mill in $\delta^{18}O$ compared to modern winter snow cover (Grinter et al., 2019; Opel et al., 2018), we suggest that modern winter precipitation in the Yana Highlands is isotopically distinctly more depleted than that at Tiksi, Zhigansk and Yakutsk.

No palaeoclimatic information based on ice-wedge isotopes is available yet from Batagay for the MIS 4 (Zyryan) and MIS 2
(Sartan) stadials. On Bol'shoy Lyakhovsky Island of the New Siberian Archipelago ice wedges from both periods show $\delta^{18}O$ values about 6‰ lower than those of MIS 3 Ice Complex ice wedges (Wetterich et al., 2011; Opel et al., 2017b; Wetterich et al., 2019). Hence, even more depleted $\delta^{18}O$ values might be expected in ice wedges of Last Glacial Maximum age at the Batagay megaslump. However, we note that distinctly depleted $\delta^{18}O$ values for MIS 4 and 2 have been found only at Bol'shoy Lyakhovsky Island.

When interpreting ice-wedge stable isotopes in terms of absolute palaeotemperatures one has to keep in mind that the isotopic composition of the oceanic moisture source has changed over glacial-interglacial timescales, with enrichment in $\delta^{18}O$ in the ocean water during past cold stages. Correcting the ice-wedge stable isotopes for this effect requires detailed knowledge about the moisture source region and in particular a better constrained chronology of ice-wedge isotopes than is available from our dataset.

In general, the inferred lower winter temperatures meet the expectations of an increased continentality during the Late Pleistocene cold stages as well as partly drier conditions indicated by the formation of composite wedges. Palaeoecological analysis (plant macrofossil remains, pollen and beetles) of Batagay megaslump deposits (Ashastina et al., 2018) indicates generally much drier conditions compared to recent times (Table 4), in particular during the cold stages. Tree and beetle indicators imply warm summers for most units, whereas macrofossil remains indicate cold to very cold winters for the Upper
Ice Complex. These patterns are generally in line with our ice-wedge interpretation and support the hypothesized higher continentality.

### 5.4 Large-scale implications for palaeoclimate and past continentality

### 5.4.1 MIS 3 interstadial wedge ice records

To assess the climate and continentality across much of west Beringia during the MIS 3 interstadial, we compare our Yana
Highlands ice-wedge stable-isotope data to a dataset of 17 other ice-wedge sites (Figure 1, Figure 7). Most of the wedges are from the Yedoma Ice Complex (Schirrmeister et al., 2011b; Schirrmeister et al., 2013), which is widely distributed in east and central Siberia. We selected all available ice wedges per study site that provide both $\delta^{18}O$ and $d$ data (Table S3). Additionally, all except two of the considered wedges or their host sediments have been directly dated by radiocarbon methods to between about 50 and 30 kyr ago (MIS 3 interstadial). We appreciate that an ice wedge may not contain a full record of this time period
and that the climate was not uniform throughout this period. It is, however, not clear yet whether millennial-scale climate oscillations known from the North Atlantic region (e.g. Dansgaard et al., 1993) also characterized the MIS 3 in east Siberia. Following Murton et al. (2017), there are hints that the MIS 3 in northern areas of northeast Siberia was a time of general

climate stability even though the interstadial climate was not monolithic. In particular, a warmer period around 40 kyr BP is well known from several palaeoecological proxies for some of the study sites (Wetterich et al., 2014) and from ice wedges of the well dated Bykovsky Peninsula section (Meyer et al., 2002a). The Batagay Upper Ice Complex ice wedge B17-IW5 shows $\delta^{18}O$ values of around –35‰, which is 6 to 2‰ lower than MIS 3 ice-wedge $\delta^{18}O$ data from northern Yakutia, the Kolyma

region and central Yakutia (Figure 8). As hypothesized, the relatively depleted $\delta^{18}O$ values from Batagay clearly point to lower winter temperatures during MIS 3 in the Yana Highlands than elsewhere, probably due to the more continental location framed by the partially glaciated Verkhoyansky and Chersky ranges reaching elevations of more than 2,000 m above sea level. Even the ice-wedge $\delta^{18}O$ data from highly continental sites in central Yakutia do not reach such low $\delta$ values as observed in the Batagay ice wedges. Interestingly, the lowest MIS 3 $\delta^{18}O$ values beside Batagay originate from the northeastern-most site,

Novaya Sibir' Island. If an Atlantic moisture source for MIS 3 wedge ice is assumed, Novaya Sibir' would represent the farthest location from the source region and therefore show strongly depleted $\delta^{18}O$ values, almost reaching those of Batagay. The peculiarity of the Yana Highlands is also apparent in the $d$ data of the MIS 3 ice wedges. The Batagay ice wedges exhibit the highest $d$ values, between 8 and 11‰, in the data set. Similar values have been found in ice wedges from central Yakutia, whereas today's coastal lowlands of northern Yakutia, the Laptev Sea islands and the Kolyma region sites show values between

2 and 7‰ (Figure 9). Again, from all northern sites, Novaya Sibir' Island shows the value closest to those of the Yana Highlands and central Yakutia.

Little is known about $d$ variation in northeast Siberian ice wedges in general, and particularly within specific time periods. Previous studies have interpreted changes in the $d$ values between the Late Pleistocene (low values) and Holocene (higher values) as changes in the moisture sources and transport pathways following the retreat of the Eurasian ice sheets (Meyer et

al., 2002a; Meyer et al., 2002b). Higher $d$ values in central Yakutia and the Verkhoyansk Mountain forelands have also been found in modern winter snow along a west-to-east transect over east Siberia from Yakutsk to Magadan (Kurita et al., 2005). This spatial pattern has been attributed to the influence of different moisture sources, with greater kinetic effects during initial evaporation, as resulting from a lower relative humidity (Kurita et al., 2005). Analogously, during MIS 3, the Yana Highlands and central Yakutia may have received precipitation from other moisture sources and/or transport pathways compared to more

northern sites. Due to their more southern location, the Yana Highlands might have received a contribution of moisture from a more southerly source, e.g. the Mediterranean. Such moisture usually has higher $d$ values compared to North Atlantic moisture (Gat et al., 2003) and may have reached this region via a different southern moisture transport pathway.

### 5.4.2 Holocene

The mean $\delta^{18}O$ values of the Holocene ice wedge sampled at the Adycha River also point to lower winter temperatures—due

to greater continentality—than other Holocene ice wedges in the data set. The Adycha samples (mean $\delta^{18}O$ –29‰) are more depleted than Holocene wedges from northeast Siberian coastal sites, which show typically mean $\delta^{18}O$ values between –27 and –23‰ (e.g. Meyer et al., 2002b; Meyer et al., 2015; Opel et al., 2017a; Wetterich et al., 2018). Holocene ice wedges from the Bykovsky Peninsula (Meyer et al., 2002a) show exceptional values (mean $\delta^{18}O$ of –28 to –26‰), which do not fit into the

regional pattern and may be related to local climate or environmental effects. In contrast, Holocene ice wedges from central Yakutia exhibit mean $\delta^{18}O$ values of –28 to –26‰ (Popp et al., 2006), which are closer to those from the Yana Highlands and also reflect the greater continentality and lower winter temperatures than the Holocene coastal sites.

The *d* values from Holocene ice wedges in Siberia are highly variable. Mean *d* values at the coastal sites are higher (compared to MIS 3), between 4 and 12‰ (e.g. Opel et al., 2011; Meyer et al., 2002b; Wetterich et al., 2008; Wetterich et al., 2018). In contrast, in the Yana Highlands the Late Holocene mean *d* value of 6‰ is lower compared to MIS 3, whereas Late Holocene *d* values in central Yakutia (between 6 and 11‰) are rather similar to MIS 3 values (Popp et al., 2006). The exceptional ice wedges from the Bykovsky Peninsula (Meyer et al., 2002a) even reach *d* values of 16‰. In summary, Holocene ice-wedge *d* values do not show a clear spatial pattern that is readily interpretable in terms of moisture sources or continentality.

## 6 Conclusions and outlook

Our stable-isotope data clearly show that winter temperatures in the Yana Highlands during MIS 3, represented by the Upper Ice Complex (Yedoma) at Batagay, and the Holocene were distinctly lower than at other ice-wedge study sites in coastal and central Yakutia. This indicates the persistence of enhanced continentality of the Yana Highlands region during at least part of the Late Pleistocene in west Beringia and during the Holocene. The stable-isotope data from narrow composite wedges of the Upper Sand unit (MIS 3–2) and an old ice wedge from the Lower Ice Complex (pre-MIS 6) are less indicative and require additional studies.

High-resolution systematic sampling and dating now needs to be carried out for all cryostratigraphic units of the Batagay permafrost sequence to validate our findings of increased continentality during MIS 3 and the Holocene, to improve the temporal resolution of the Batagay ice-wedge record, and to elucidate the palaeoclimatic history from other time slices. Of particular interest are the lower and uppermost parts of the Upper Ice Complex, likely representing MIS 4 and 2 stadials, as well as more detailed studies of the yet undated Lower Ice Complex. To establish reliable ice-wedge chronologies for the Upper Ice Complex and Upper Sand unit, radiocarbon dating should include macro remains, dissolved organic carbon and $CO_2$ from gas bubbles (Lachniet et al., 2012; Kim et al., 2019). Additionally, previous luminescence dating results (Ashastina et al., 2017) need to be validated.

**Author contributions**

Thomas Opel initiated and designed the present study and wrote the paper with contributions by the other co-authors. Thomas Opel, Julian Murton, and Kseniia Ashastina sampled and described ground ice and host sediments, supported by Petr Danilov and Vasily Boeskorov. Hanno Meyer carried out stable-isotope analyses and supported the interpretation. Hendrik Grotheer and Gesine Mollenhauer conducted the radiocarbon dating. Frank Günther provided GIS analysis and maps. Sebastian

Wetterich and Lutz Schirrmeister collected ice-wedge data for comparison and supported data analysis and interpretation. All co-authors contributed to the final discussion of the results and interpretations.

## Competing interests

The authors declare that they have no conflict of interest.

## Acknowledgements

We would like to thank Erel Strutchkov for support of fieldwork as well as Luidmila Pestryakova and Waldemar Schneider for organisation of export of samples. Mikaela Weiner supported stable-isotope analysis. Thomas Opel and Sebastian Wetterich acknowledge funding from German Research Foundation (DFG grants OP217/3-1, OP217/4-1 and WE4390/7-1, respectively). Frank Günther was supported by ERC #338335 and by DAAD with funds from BMBF and the EU's Marie Curie
Actions Programme, REA grant agreement #605728 (P.R.I.M.E.). We acknowledge three anonymous referees and the editor Denis-Didier Rousseau for constructive comments, which helped us to improve the manuscript.

## Data availability

The new ice-wedge and new $^{14}$C data presented in this paper as well as the ice-wedge data used for spatial comparisons will be made available at https://www.pangaea.de after acceptance of this paper.

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

**Table 1. Cryostratigraphic units of the Batagay megaslump as presented by Ashastina et al. (2017) and Murton et al. (2017) and the reconciled stratigraphic framework used in this study and proposed for future studies.**

| Ashastina et al. (2017) | | Murton et al. (2017) | | This study |
|---|---|---|---|---|
| Unit | Description | Unit | Description | Unit |
| Unit I | Active layer | Unit 6 | Near-surface sand | Near-surface layer |
| Unit II | Yedoma Ice Complex | Unit 5 | Upper sand | Upper Sand |
| | | Unit 4 | Upper Ice Complex | Upper Ice Complex |
| Unit III | Organic-rich layer | Woody lens | Woody lens | Wood and plant remains |
| Unit IV | Layered brown sands | Unit 3 | Lower sand | Lower Sand |
| Unit V | Ancient Ice Complex | Unit 2 | Lower Ice Complex | Lower Ice Complex |
| - | - | Unit 1 | Diamicton | Diamicton |

**Table 2. Stable isotope ($\delta^{18}$O, $\delta$D and *d*) minimum, mean and maximum values, standard deviations as well as slopes and intercept in the $\delta^{18}$O–$\delta$D diagram for all ice and composite wedges, intrasedimental ice, outflow water and rain water, respectively.**

| Ice wedge/ Type of water | Width (m) | Samples (n) | $\delta^{18}$O min (‰) | $\delta^{18}$O mean (‰) | $\delta^{18}$O max (‰) | $\delta^{18}$O sd (‰) | $\delta$D min (‰) | $\delta$D mean (‰) | $\delta$D max (‰) | $\delta$D sd (‰) | *d* min (‰) | *d* mean (‰) | *d* max (‰) | *d* sd (‰) | Slope | Intercept | $R^2$ |
|---|---|---|---|---|---|---|---|---|---|---|---|---|---|---|---|---|---|
| **Batagay megaslump** | | | | | | | | | | | | | | | | | |
| *Upper Sand unit* | | | | | | | | | | | | | | | | | |
| B17-IW2 (CW) | 0.25 | 4 | -33.04 | -32.83 | -32.47 | 0.25 | -250.2 | -247.5 | -242.8 | 3.3 | 13.7 | 15.1 | 16.9 | 1.5 | 12.51 | 163.29 | 0.92 |
| B17-IW3 (CW) | 0.2 | 2 | -32.14 | -31.87 | -31.60 | 0.38 | -244.2 | -240.3 | -236.4 | 5.5 | 12.9 | 14.7 | 16.4 | 2.4 | 14.47 | 220.91 | 1.00 |
| B17-IW4 | 0.4 | 6 | -30.10 | -29.93 | -29.71 | 0.13 | -229.6 | -227.8 | -223.9 | 2.2 | 10.0 | 11.6 | 13.8 | 1.3 | 15.03 | 222.01 | 0.82 |
| Intrased. ice | n/a | 3 | -32.61 | -30.55 | -29.21 | 1.82 | -242.2 | -220.0 | -207.3 | 19.3 | 18.7 | 24.4 | 28.2 | 5.1 | 10.61 | 104.19 | 0.99 |
| *Upper Ice Complex* | | | | | | | | | | | | | | | | | |
| B17-IW5 | 1.6 | 12 | -36.17 | -34.89 | -33.50 | 0.86 | -280.6 | -271.0 | -260.8 | 6.4 | 6.8 | 8.1 | 10.3 | 0.9 | 7.41 | -12.36 | 0.99 |
| B17-IW6 | 0.5 | 4 | -35.52 | -35.05 | -34.27 | 0.57 | -274.9 | -269.4 | -259.7 | 7.0 | 8.5 | 11.0 | 14.4 | 2.5 | 12.25 | 159.79 | 0.99 |
| Intrased. ice | n/a | 2 | -27.63 | -26.19 | -24.75 | 2.04 | -198.5 | -195.0 | -191.5 | 5.0 | 6.5 | 14.5 | 22.5 | 11.3 | 2.44 | -131.03 | 1.00 |
| *Lower Sand unit* | | | | | | | | | | | | | | | | | |
| Intrased. ice | n/a | 2 | -32.31 | -32.28 | -32.24 | 0.05 | -241.6 | -241.6 | -241.6 | 0.1 | 16.4 | 16.6 | 16.8 | 0.3 | 1.33 | -198.54 | 1.00 |
| *Lower Ice Complex* | | | | | | | | | | | | | | | | | |
| B17-IW1 | 0.5 | 6 | -33.18 | -33.09 | -32.91 | 0.10 | -258.3 | -256.7 | -254.8 | 1.5 | 6.9 | 8.0 | 9.4 | 0.8 | 12.78 | 166.23 | 0.80 |
| Intrased. ice | n/a | 2 | -32.56 | -32.48 | -32.39 | 0.13 | -247.4 | -247.1 | -246.7 | 0.5 | 12.4 | 12.8 | 13.1 | 0.5 | 4.09 | -114.27 | 1.00 |
| *Other* | | | | | | | | | | | | | | | | | |

| | | | | | | | | | | | | | | | | | |
|---|---|---|---|---|---|---|---|---|---|---|---|---|---|---|---|---|---|
| Outflow water | n/a | 7 | -32.29 | -31.73 | -31.35 | 0.34 | -244.0 | -238.6 | -235.4 | 3.1 | 14.3 | 15.2 | 16.1 | 0.6 | 8.84 | 41.83 | 0.97 |
| Rain water | n/a | 4 | -16.82 | -14.33 | -10.17 | 2.92 | -131.7 | 112.0 | -79.0 | 23.1 | 2.3 | 2.6 | 2.9 | 0.2 | 7.93 | 1.65 | 1.00 |
| **Adycha River** | | | | | | | | | | | | | | | | | |
| *Holocene river bank Adycha River* | | | | | | | | | | | | | | | | | |
| A17-IW3 | 0.3 | 10 | -30.40 | -28.98 | -28.15 | 0.74 | -236.6 | -226.3 | -220.5 | 5.2 | 4.4 | 5.5 | 7.3 | 0.9 | 7.01 | -23.25 | 0.99 |
| **Ice-wedge cluster** | | | | | | | | | | | | | | | | | |
| Cluster 1 | (B17-IW1, B17-IW5, B17-IW6, A17-IW3) | | | | | | | | | | | | | | 7.45 | -10.46 | 1.00 |
| Cluster 2 | (B17-IW2, B17-IW3, B17-IW4) | | | | | | | | | | | | | | 6.88 | -21.72 | 0.97 |

**Table 3. Radiocarbon ages of organic remains in ice wedges (sample ID includes IW) and host sediments of the Batagay megaslump and at the Adycha River. The samples were radiocarbon dated as gas targets (Lab ID ends with 1.1) and graphite targets (Lab ID ends with 2.1).**

| Sample ID | Depth (m bs) | Lab ID | $F^{14}C$ | Radiocarbon age (yr BP) | Calibrated age 95.4% (cal yr b2k) | Dated material | Remarks |
|---|---|---|---|---|---|---|---|
| **Batagay megaslump** | | | | | | | |
| *Upper Sand unit* | | | | | | | |
| B17-S2-AMS4-1 | 11 | 1689.1.1 | <0.0094 | >37,500 | n/a | *Empetrum nigrum* leaf, twigs, Cyperaceae remains (sediment) | |
| | | 1689.2.1 | 0.0084±0.0002 | 38,348±236 | 42,863–42,140 | | |
| B17-S2-AMS4-2 | 11 | 1690.1.1 | <0.0094 | >37,500 | n/a | Cyperaceae stems and roots (sediment) | |
| *Upper Ice Complex* | | | | | | | |
| B17-IW5-02 | 20.3 | 1684.1.1 | 0.0453±0.0030 | 24,858±536 | 30,355–27,926 | unidentified plant (twigs, roots and florets) remains (ice wedge) | |
| B17-IW6-04 | 25.8 | 1686.1.1 | <0.0094 | >37,500 | n/a | unidentified plant (bract fragments and roots) and insect (complete pieces and fragments of elytron) (ice wedge) | |
| B17-S2-AMS5 | 20.3 | 1691.1.1 | <0.0094 | >37,500 | n/a | leaf fragment, roots (sediment) | |
| | | 1691.2.1 | 0.0027±0.0002 | 47,550±677 | >50,033 | | |
| B17-S2-AMS6 | 25.8 | 1692.1.1 | <0.0094 | >37,500 | n/a | Cyperaceae remains, roots (sediment) | |
| *Lower Ice Complex* | | | | | | | |
| B17-IW1-04 | ~50 | 1683.1.1 | 0.1400±0.0062 | 15,792±358 | 20,044–18,402 | hare dropping (ice wedge) | Likely redistributed |
| B17-S1-AMS2 | ~49.5 | 1688.1.1 | <0.0094 | >37,500 | n/a | twigs, Cyperaceae stems and roots (sediment) | |
| | | 1688.2.1 | <0.0013 | >53,400 | n/a | | |
| **Adycha River** | | | | | | | |
| *Holocene river bank Adycha River* | | | | | | | |
| A17-K-03 | 2 | 1687.1.1 | 0.9738±0.0132 | 213±109 | <514 | rootlets and *Larix* needle (sediment) | |

**Table 4. Summary of past climate in the Yana Highlands. Palaeoecological interpretations are from (Ashastina et al., 2018) and based on fieldwork in 2014.**

| Unit<br>Ice wedge<br>Chronology | Cryostratigraphy | Interpretation | IW isotopes | Interpretation | Palaeoecological interpretation |
|---|---|---|---|---|---|
| Adycha, river bank<br>Ice wedge A17-IW3<br>Late Holocene to recent | Narrow ice wedge, little sediment | Moist (high melt water supply) | $\delta^{18}O$: –29‰<br>$d$: 7‰ | Cold and rather moist winters | - |
| Batagay, Upper Sand unit<br>Composite wedges<br>B17-IW2–3<br>MIS 3–2 | Narrow composite wedges | Dry (little snow melt water) and/or high sediment supply | $\delta^{18}O$: –33‰, –32‰<br>$d$: 15‰ | Dry winters colder than in the Holocene | Dry summers, $T_{july}$ >12°C (trees) |
| Batagay, Upper Sand unit<br>Ice wedge B17-IW4<br>MIS 3–2 | Narrow ice wedge, little to medium sediment content | Rather moist (higher melt water supply compared to composite wedges) | $\delta^{18}O$: –30‰<br>$d$: 12‰ | Cold and rather dry winters (Epigenetic Holocene IW?) | Dry summers, $T_{july}$ >12°C (trees) |
| Batagay, Upper Ice Complex<br>Ice wedge B17-IW5<br>MIS 3 | Wide ice wedge, little to medium sediment content | (rather) moist (high melt water supply) | $\delta^{18}O$: –35‰<br>$d$: 8‰ | Coldest and rather moist winters | Dry and (very) warm summers, very cold and dry winters (beetles) |
| Batagay, Upper Ice Complex<br>Ice wedge B17-IW6<br>MIS 3 | Medium ice wedge, high sediment content | Rather dry (drier than IW5, moister than IWs 2/3) | $\delta^{18}O$: –35‰<br>$d$: 11‰ | Coldest and rather dry winters | Dry summers, cold winters |
| Batagay, Lower Ice Complex<br>Ice wedge B17-IW1<br>Pre-MIS 6 | Narrow ice wedge, little sediment content | Moist, high melt water supply | $\delta^{18}O$: –33‰<br>$d$: 8‰ | Winters warmer than in MIS 3 and rather moist | - |

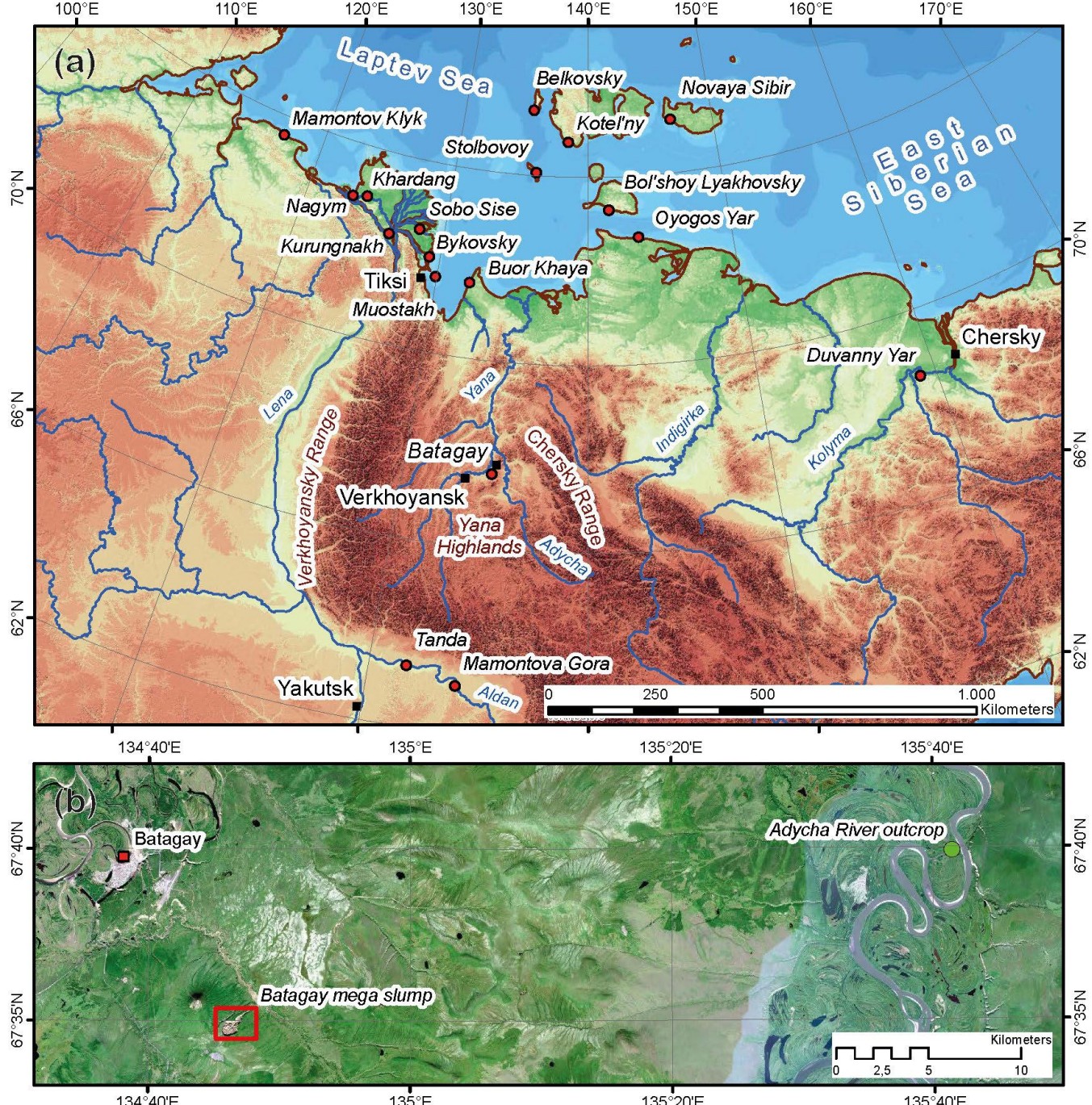

**Figure 1. (top) Location map of study region in northeast Siberia showing all ice-wedge study sites mentioned. (bottom) Study sites in the Yana Highlands (Background image: Sentinel-2 mosaic, August 7, August 21, 2017).**

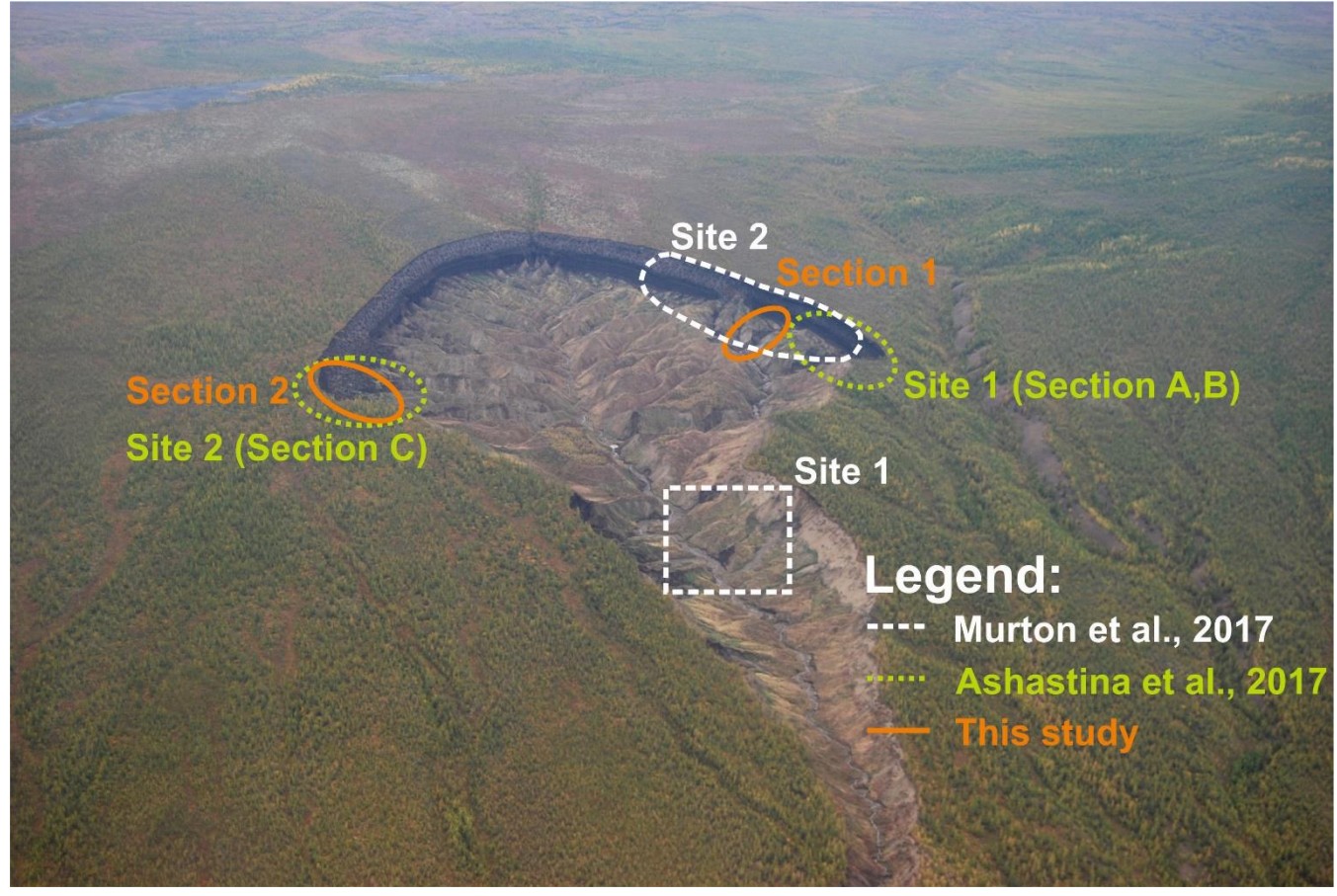

**Figure 2. Overview photograph of the Batagay megaslump showing the study sites of (Ashastina et al., 2017; Murton et al., 2017) and this study. Photograph taken by Alexander Gabyshev in 2015.**

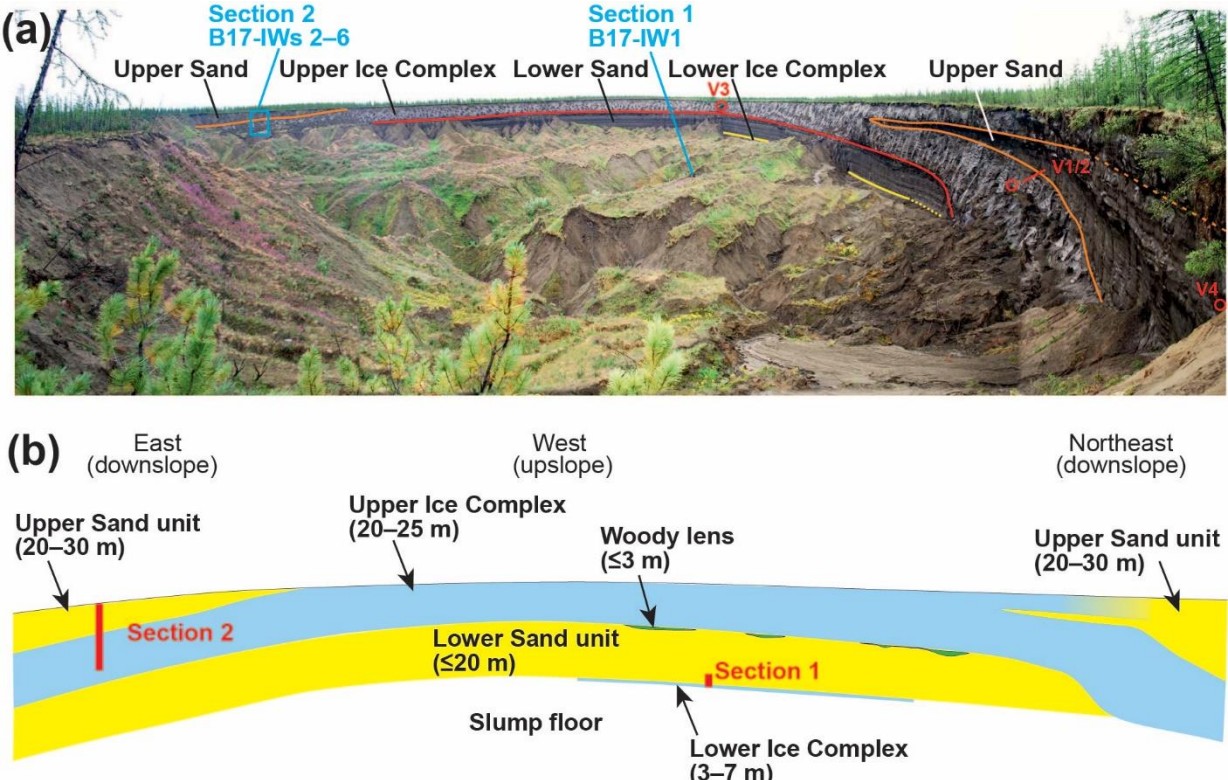

Figure 3. (a) Panoramic photograph of the Batagay megaslump with main stratigraphic units and approximate sampling locations of ice wedges (blue arrows). Red circles and numbers indicate the approximate sampling locations of ice wedges used by (Vasil'chuk et al., 2017), (b) Schematic stratigraphic section of the exposed units along the headwall of the Batagay megaslump with approximate positions of the studied sections.

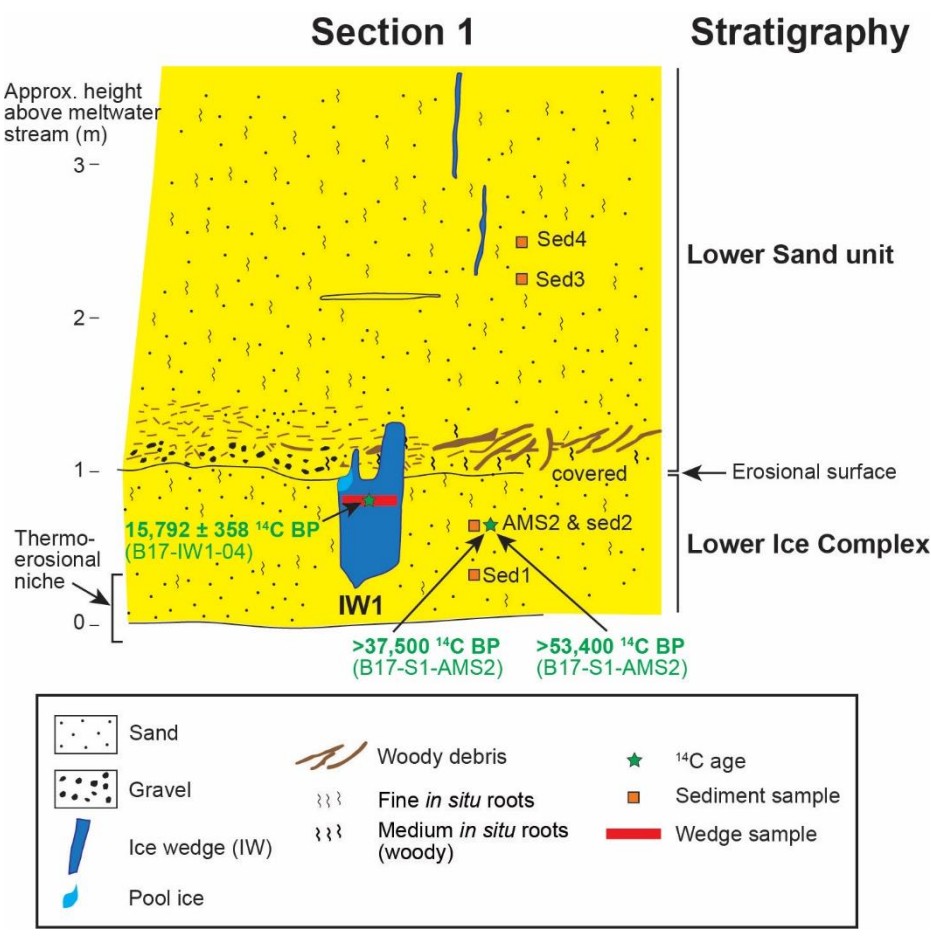

**Figure 4. Schematic cryostratigraphic sketch of Section 1, Batagay megaslump.**

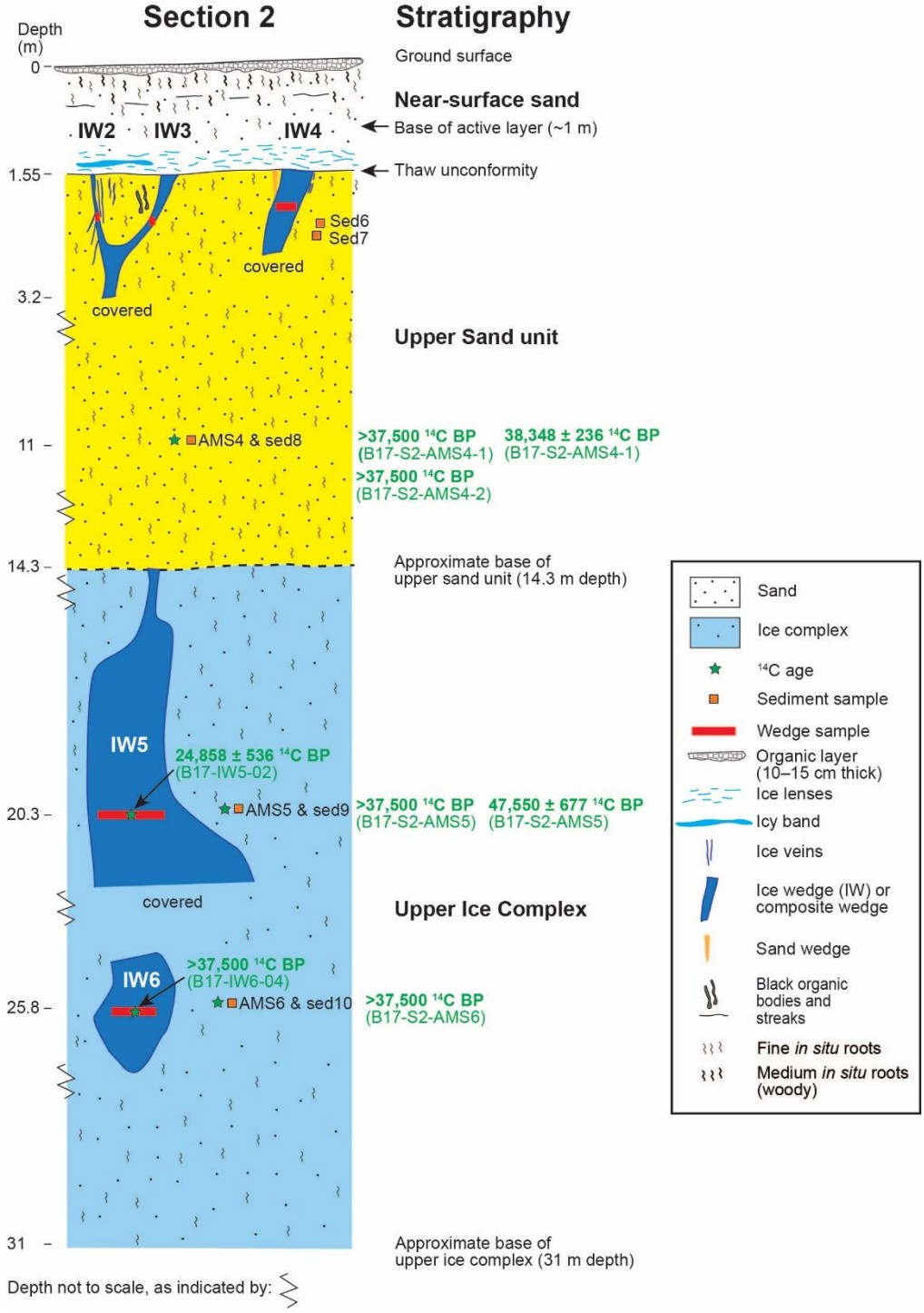

**Figure 5. Schematic cryostratigraphic sketch of Section 2, Batagay megaslump.**

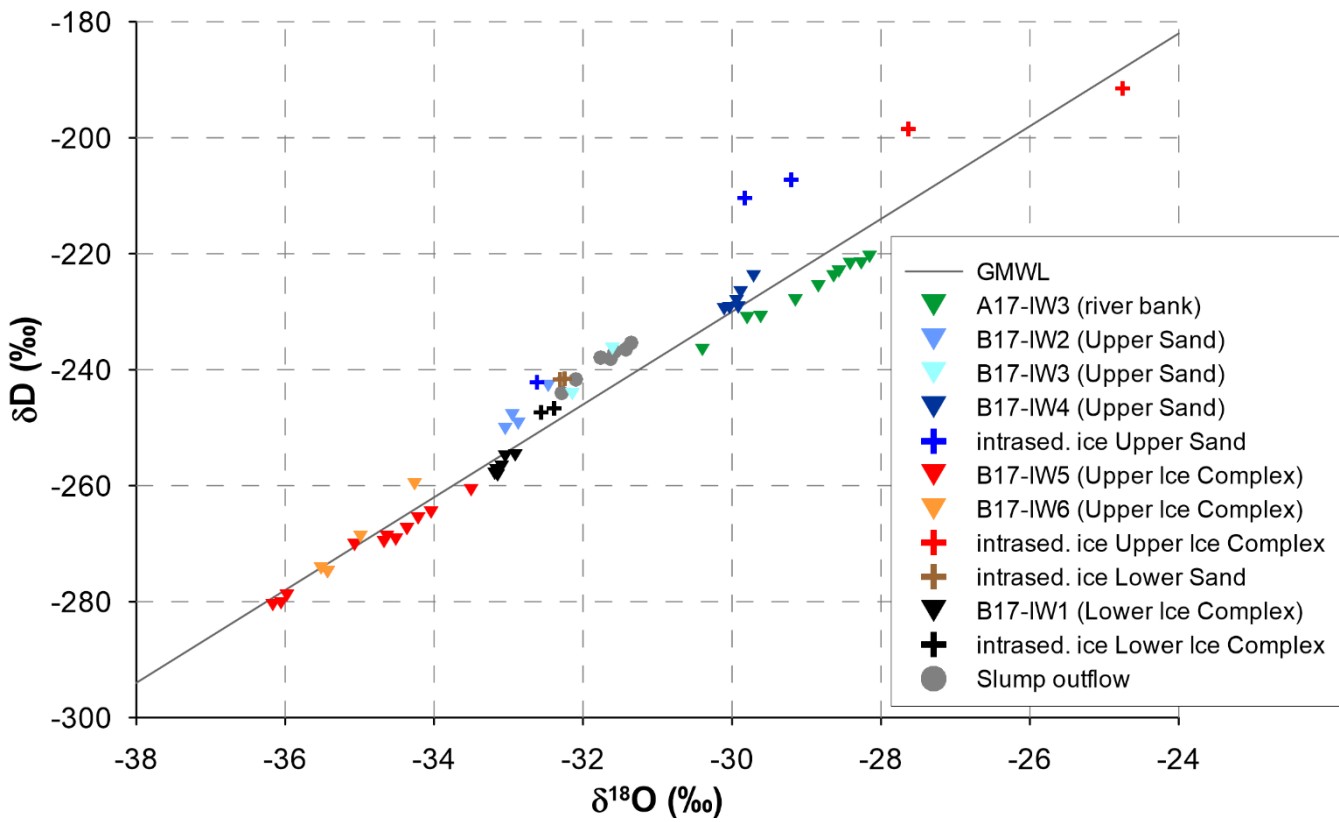

Figure 6. Co-isotopic δ¹⁸O–δD diagram of ice wedges and intrasedimental ice sampled at the Batagay megaslump and the Adycha River. The data from the Batagay megaslump outflow samples record an integrated signal of the entire Batagay permafrost sequence and modern precipitation.

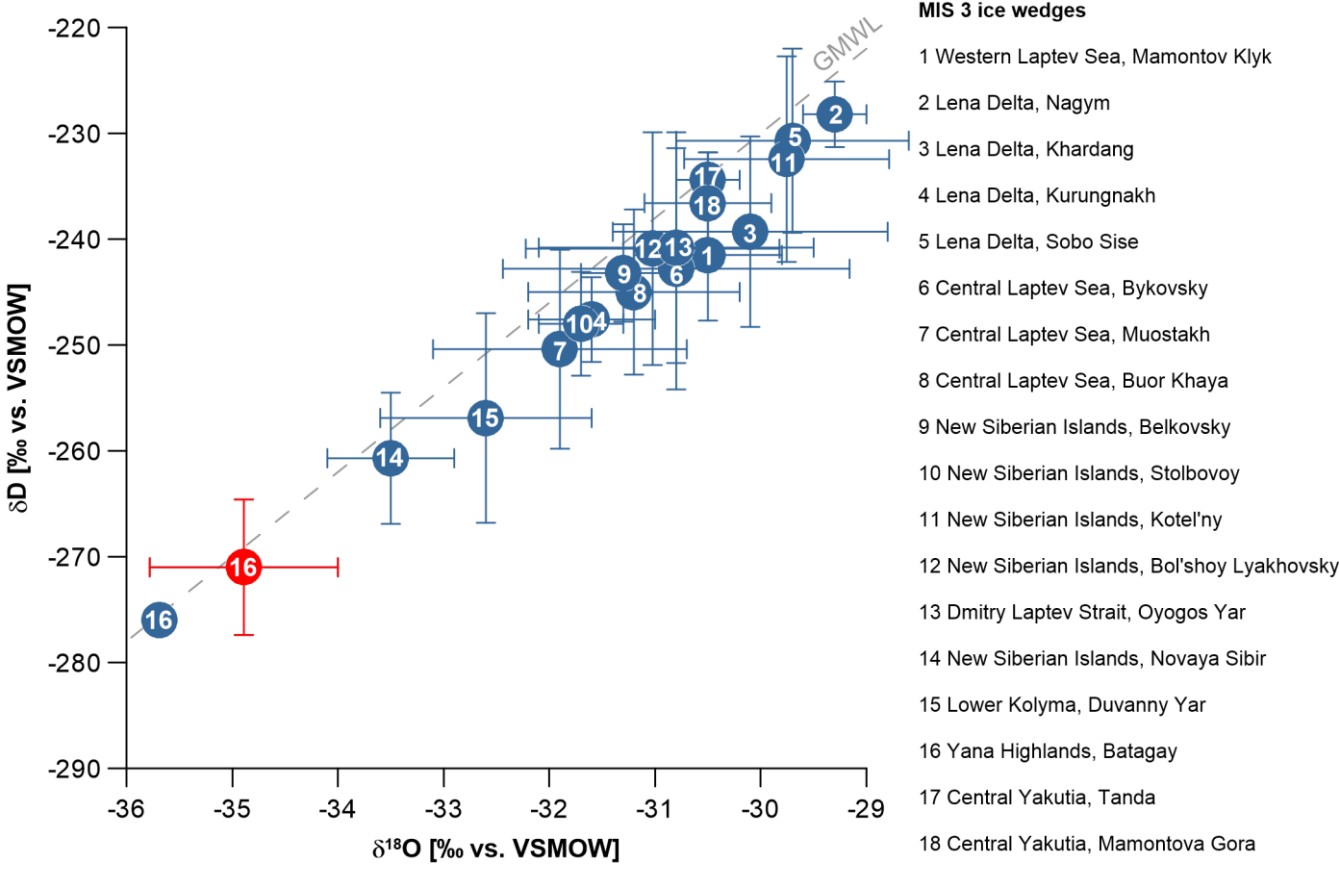

**Figure 7. Comparison of stable isotope data from Siberian ice wedges attributed to the MIS 3 interstadial (about 50 to 30 kyr), comprising data from the western Laptev Sea (Magens, 2005), the Lena River Delta (Schirrmeister et al., 2003; Schirrmeister et al., 2011a; Wetterich et al., 2008; Opel, unpublished), the central Laptev Sea (Meyer et al., 2002a; Meyer/Opel, unpublished; Schirrmeister et al., 2017), the New Siberian Islands and the Dmitry Laptev Strait (Schirrmeister, unpublished; Wetterich et al., 2014; Opel et al., 2017b), the Kolyma Lowland (Strauss, 2010), the Yana Highlands (this study; Vasil'chuk et al., 2017), and central Yakutia (Schirrmeister, unpublished; Popp et al., 2006). The new data from Batagay are marked in red. Further information is given in Table S3.**

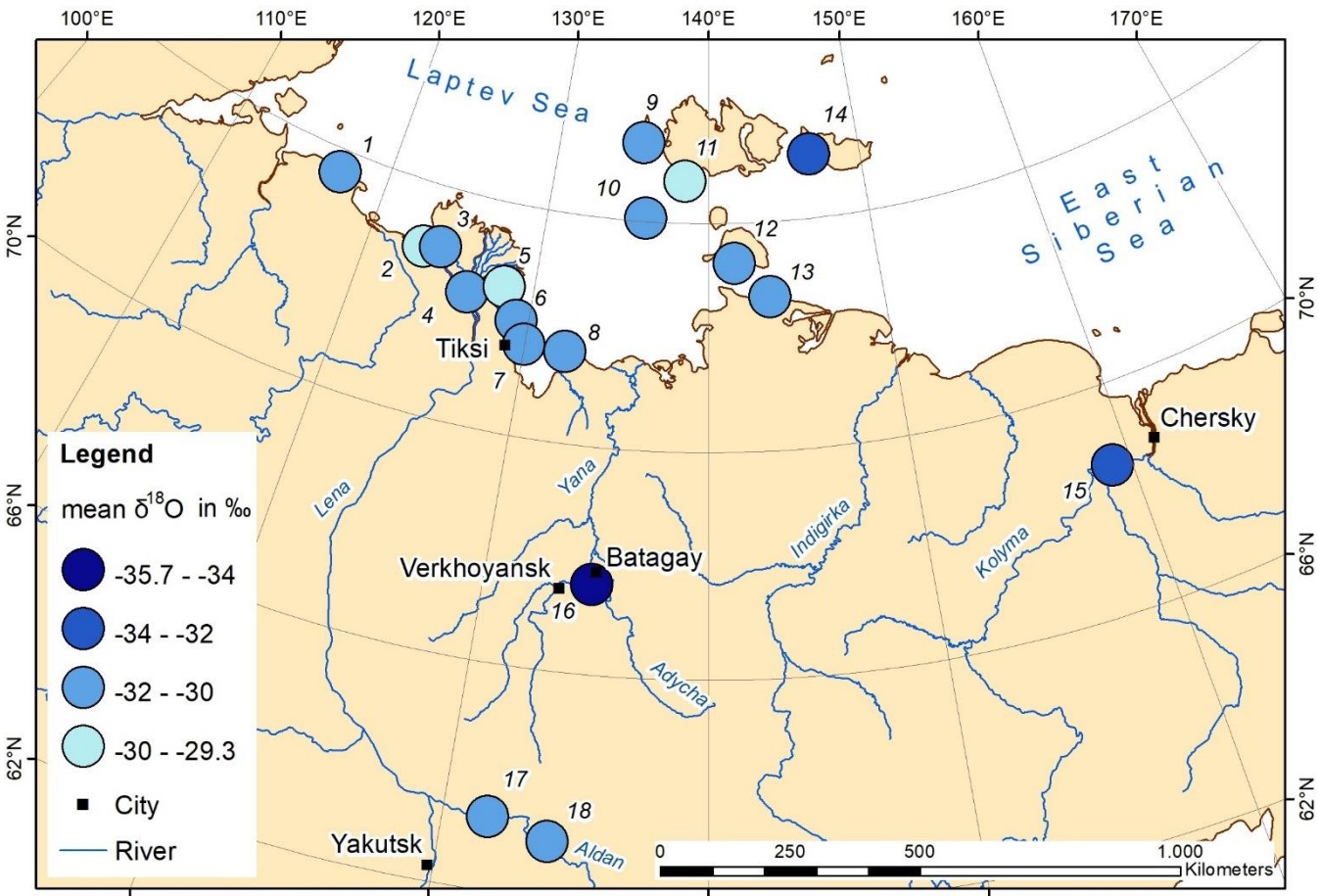

**Figure 8. Map of mean δ¹⁸O data from Siberian ice wedges attributed to the MIS 3 interstadial (about 50 to 30 kyr), Site IDs are given in Figure 7 and further details in Table S3.**

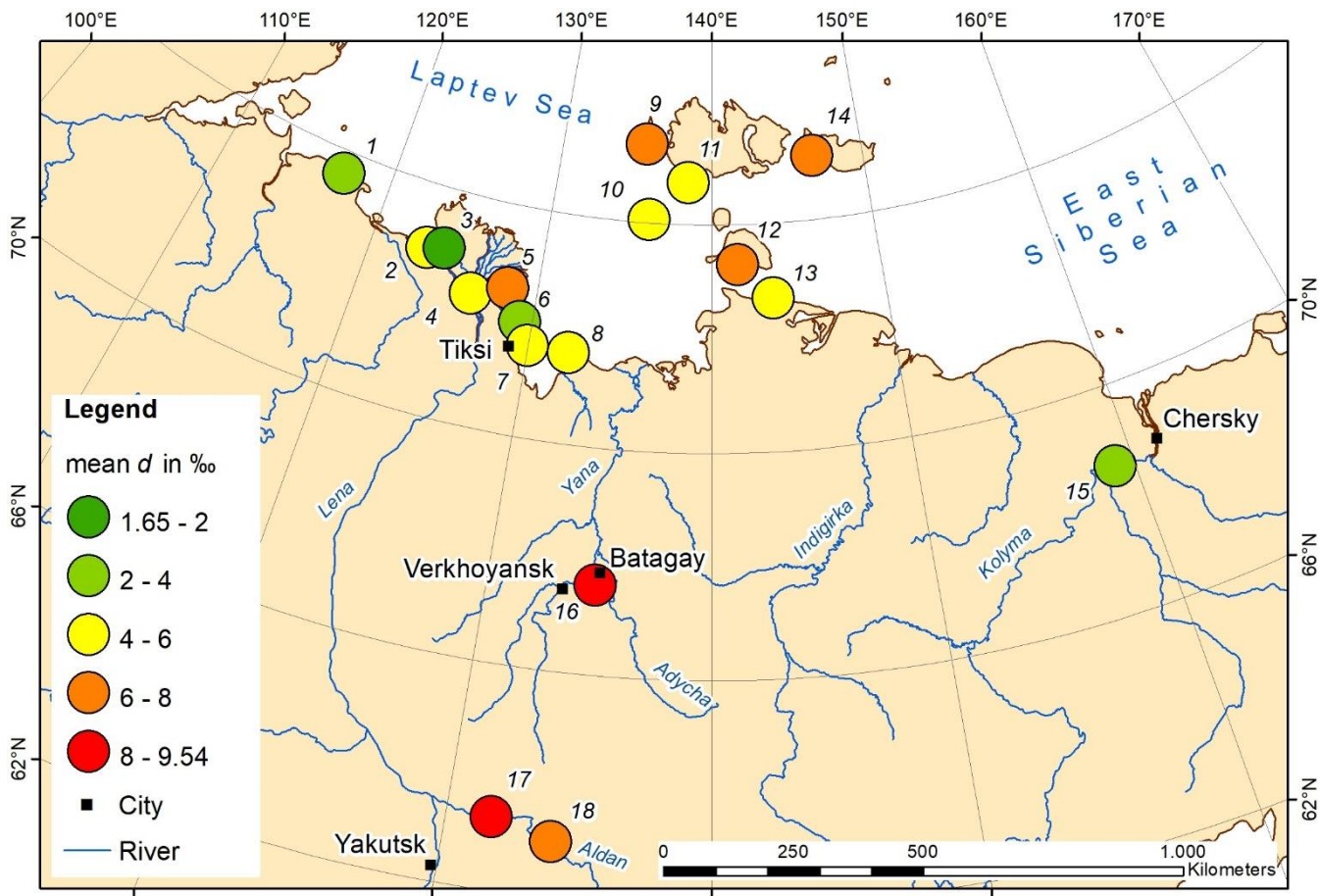

**Figure 9.** Map of mean *d* data from Siberian ice wedges attributed to the MIS 3 interstadial (about 50 to 30 kyr), Site IDs are given in Figure 7 and further details in Table S3.