# Peer review of "Past climate and continentality inferred from ice wedges at Batagay megaslump in the Northern Hemisphere's most continental region, Yana Highlands, interior Yakutia"

_Climate of the Past, 2018_

## Referee Comment (RC1) · Anonymous Referee #1 · 24 Dec 2018

The authors present a reconstruction of winter precipitation isotope ratios and inferred winter temperatures from relict ice wedges in Pleistocene strata in the Batagay megaslump headwall and late Holocene wedge ice near the Adycha River. These wedges span the last ∼140 ka. The authors claim this study fills an important gap in knowledge of paleoclimate of the Yana Highlands. Temporal variations in the isotopic composition of the ice wedges, particularly the wedges that are well-dated and substantial in size (i.e., least prone to post-depositional overprinting), broadly support

the conclusion that past stadial and interstadial winters in this region were cooler than today. Further, wedge ice from other areas across Siberia are more enriched than the Yana Highlands, both today and in the past, which suggests this region has always been the most continental area in northern Siberia.

I read this paper with great interest. Relict ice wedges are an important paleo-meteoric water archive with a tightly constrained seasonality (winter only) and offer some of the oldest known ice in the Northern Hemisphere, extending beyond the temporal limits of the Greenland ice cores. As I see it, ice wedges have an important role to play in advancing our knowledge of Quaternary climate change in non-glaciated Arctic regions where ice cores are not possible.

The authors do a good job explaining their methods and summarising the most important features of this interesting dataset. There are several uncertainties about the integrity of the smaller ice wedge samples, and dating of the pre-MIS 3 wedges, but the authors are up-front about these uncertainties and to a large extent they do not confound the conclusions highlighted in this paper. This paper lays the groundwork for future studies to develop more detailed ice wedge records and paleoclimate inferences from this site. The topic and scope of this work are highly appropriate for Climate of the Past.

I am mostly satisfied with the paper in its current form, but I have a few concerns that I feel should be addressed before it is accepted in final form. Following these revisions I would recommend this paper is accepted.

Major comments: To establish how much more continental the study area was in the past, the authors should consider that the global oceans were more enriched during past cold stages. For example, during MIS2 mean ocean water was $\sim 1.2‰$ enriched in 18O. In the discussion, please acknowledge this and provide some discussion – note that standardising for this effect would make some of the relict wedge ice (e.g., B17-IW4) similar in value to the late Holocene wedge ice. I do appreciate there are uncertainties about paleo-source region and possible heterogeneity in the isotope composition of marine source waters. However, some attention to this issue is needed.

specific comments P2, L31 – 'right side' is ambiguous, since it depends on which direction one is oriented. P3, L13 – 'MIS6 or MIS7'. Does cryostratigraphy provide any indication if this deposit (and the ice wedges) represent a glacial or interglacial period? Ideally say which is more likely. P4, L1 – you refer to this as the second study site, which is understandable but also confusing since there is a 'Site 2' in Figure 2. P4, L23-25 – if possible, please give a reference for pore ice-wedge ice exchange. P5, L21 – 'outlined below' P6, L23 – what is meant by redistribution? P11, L5-8 – The co-isotope linearity of this cluster is remarkable. Why not report the slope, intercept and r2 for a line drawn through all data within a cluster, as was done for the individual wedges? P11, L8-9 – arguably, the MIS6 wedge is part of the first cluster. Only one of the datapoints is an outlier. P11, L12-15 – this could be explained more. If you are correct, then divergence from the Cluster 1 line could provide valuable information about aridity. P11, L17 – please add a reference (e.g., Pfahl and Sodeman, 2014, Clim. Past.) P11, L21 – this is an interesting line of discussion. Can you expand on this point, and comment on how different snowpack evolutions would be expected to influence the isotopic composition of the eventual meltwaters? P12, L2-5, the point of this sentence is not entirely clear. P12, L9, specify that you are talking about d18O. Also, I would advise against specifying a number (e.g., -40). . .better to simply say even more depleted values compared to MIS3 wedge ice is expected. P12, L18-19, please clarify what is meant here. P14, L1, clarify that mean dexcess of 6 permille is a Late Holocene value. . .same for the Yakutia example. P14, L18-20, this last sentence seems irrelevant to the study. . .further, it is not clear how this study area provides the ideal conditions are validating/advancing the Cl dating method. Suggest deleting this sentence. Figure 7. the #7 datapoint is missing x/y error bars.

---

## Referee Comment (RC2) · Anonymous Referee #2 · 27 Dec 2018

The manuscript presents stratigraphy, geochronology, and ice-wedge stable isotope data from the Batagay megaslump- a remarkable bluff of late-Middle- and and Late-Pleistocene sediments in interior Yakutia exposed by a spectacularly-large thaw slump. They add some new radiocarbon dates to the emerging chronological framework for this site, and provide some new stable O and H isotope data for a small number of ice-wedges ranging in age from MIS 6(?) to the Holocene. Focusing on the broad MIS 3 interval, the authors conclude that winter temperatures during MIS 3 were colder at

this site in interior Yakutia, relative to a compilation of purportedly MIS 3 ice wedges from mostly coastal sites.

In principle I think data and discussion from this kind of proxy archive are a good fit for the scope and audience for Climate of the Past. I also think this site is really a remarkable find, particularly because of the potential for preservation of pre-MIS 5 relict ground ice. The writing and figures are mostly clear. But several factors make me unable to recommend publication: 1 Relatively low number of analyses 2 Poor chronology that inhibits meaningful comparison between sites 3 Speculative nature of the paleoclimate discussion.

I elaborate on these points below, with more specific comments at the end.

Note: Though the title of the manuscript alludes to Middle and Late Pleistocene climate and continentality, my main points of concern are limited to the MIS 3 part of the story because the authors acknowledge that the data from younger and older parts of the sequence are equivocal (p14/line10).

1. The Batagay megaslump headwall exposure is over 1 km long, yet the conclusions re: MIS 3 rest on data collected from only two ice wedges at a single measured section. The discussion and conclusion around Holocene climate is similarly based on analyses of only a single ice wedge. I realize that field work on sites like this is difficult and potentially dangerous, but the chainsaw sampling is rapid and contemporary analytical techniques allow for hundreds of samples to be analyzed in relatively short order. Rather than limited and equivocal results from a reconnaissance visit to the field site, my sense is that this topic deserves "high-resolution systematic sampling and dating", as the authors point out in their conclusion.

2. Most critically, I question if the available data support a meaningful conclusion re: MIS 3. In other words, what does it actually mean to compare a single probably-MIS3 ice-wedge from one site to another single probably-MIS3 ice-wedge from another site (as is done in Fig 7 and Table S2), since this interval spans ∼30,000 years and includes

some pretty high-amplitude multi-millennial-scale climate oscillations in high-resolution proxy records? The time interval is also notoriously difficult to date accurately with 14C methods, and many purportedly finite ∼35-45 14C ka BP dates in the literature ought to be viewed with a strong dose of skepticism (notably, for example, the purported MIS 3 chronology for Mamontova Gora - an important comparative site Fig 7/Table S2). Seven out of the 18 sites in the Fig 7/Table S2 ice-wedge compilation are unpublished, so readers can't assess the reliability of these chronologies for themselves. The authors mention the issue of dating and a long MIS 3 (p12/line26) but do not really address it in a way that justifies the approach. One example of the interpretive difficulties: Novaya Sibir, Belkovsky, and Kotel'ny are all above 74 degN in the New Siberian Islands, yet only Novaya Sibir has relatively depleted isotope composition. Is the between-site difference in isotope composition due to differing age or some site-specific factor? Either way, the lack of good chronological control inhibits meaningful comparison.

One last point of criticism on the comparison of different sites: why was the compilation/comparison limited to just one "MIS 3" ice-wedge from each site? In the context of this analysis, would it not be more useful to compare the average isotope composition of multiple ice-wedges from a particular stratigraphic interval (e.g. the 10 ice wedges with dD and d18O data attributed to the yedoma ice complex in Opel et al 2017)?

3. I acknowledge that quantitative paleoclimate reconstruction from this type of archive is highly uncertain, but the climate implications presented here are vague. Differences in isotope composition between areas are quantified, but then unsupported paleotemperature interpreations are made (e.g. "significantly lower [temperatures]" p14/line7; "extremely low winter temperatures" p11/line23 vs "very low winter temperatures" p11/line26). These distinctions need to be defined.

The authors have not really addressed the issue of paleogeography, nor potential differences in moisture source both through time and for different sites. For example (assuming for a moment that it's possible to meaningfully compare MIS 3 IWs at the Batagay site to those compiled in Fig 7), there's an interesting spatial pattern whereby the Novaya Sibir Island site also has highly depleted IW isotope composition during MIS 3. What are the paleogeographic implications of Late Pleistocene sea level change, with respect to continentality? What are current and modelled MIS 3 moisture sources for that site and the Batagay site? Is there paleoceanographic proxy evidence (e.g. from planktic forams) for changes in surface water isotopic composition at likely moisture sources? All of these points would likely provide useful context for evaluating the data presented in the manuscript.

Other points: This manuscript, which includes many co-authors on earlier papers that document the chronostratigraphic framework for the site, introduces yet another unit-stratigraphic nomenclature for the Batagay megaslump headwall exposure. For example, at least by my interpretation, "upper Ice Complex" (this ms) = Unit III (Ashastina et al. 2017) = Unit 4 (Murton et al. 2017). Given the potential importance of this site, and since all the key players are co-authors on this manuscript, it would be very useful to the community if the authors could reconcile these different frameworks here in this manuscript.

This group is highly experienced in stable isotope studies of ground ice. Nevertheless, it would be useful to provide some additional methods data. Is the quoted lab precision for dD and d18O 1sigma or 2sigma? What are the summary statistics for the internal quality control secondary standard? And most importantly (p4/line24), how exactly did the authors decide which samples to exclude from further analyses? The description in the manuscript is very vague (please clarify, with a citation or two, what exchange processes are being invoked between IW and pore ice), and I would strongly prefer that the authors present ALL the ice-wedge stable isotope data first and then justify to readers why some data should be excluded from further consideration.

Specific comments:

Referencing: There are points in the introduction and discussion where it would be

useful and appropriate to include some citations to relevant work in North America, where there is a long tradition of stable isotope work on ice wedges (e.g. Fraser and Burn; Michel) and the stratigraphic complexities of Middle/Late Pleistocene permafrost exposures (e.g. Péwé, Westgate; Fraser & Burn; Froese, Reyes).

Title: Given the substantial interpretive and chronological uncertainty re: the lower sand and upper unit ice wedge data, I suggest removing "Middle and Late Pleistocene" from the title and replacing it with something more specific

Section 1 in the slump floor: Why is the one sampled wedge from the lower sand collected away from the exposed headwall, as indicated in Fig 3? Can authors reject the possibility that the sampled section is actually younger material displaced into an apparently lower stratigraphic position by slumping? And I'm troubled by the rejection of the 14C age on hare droppings from the ice-wedge. If this was a pristine, freshly-exposed ice-wedge, how would younger material be incorporated into the wedge itself? Surely the outermost surface of the wedge ice is removed prior to sampling? And if material "entered into or later froze onto the surface" (p8/line24), doesn't this also imply possible reliability issues with the isotope data from that wedge?

p8/line18: Do you mean ". . .different stratigraphic contexts"? This would make more sense. Also, the next few sentences of this paragraph are pretty vague and not particularly useful. I think it's pretty obvious now that adequate dating of these sediments is going to be a major challenge. There's some mention of alternative approaches in the conclusion section, which really should be moved into the discussion and properly addressed.

wood layer and the thaw unconformity (p. 8/9 transition): I assume you mean ". . .situated above the lower sand unit and BELOW the upper Ice Complex. . .." on p8/line29, since you reasonably attribute the wood layer to the last interglacial?

14C dating (Table 2 and p5): I appreciate the details on pre-treatement and analysis. Please clarify if smaller blanks were measured for background correction of the many

samples with low mass of organic C (Table 2).

Fig. 5: The changing vertical scale is confusing. Since the ice-wedge morphology is important in this context, I recommend a stratigraphic diagram to scale, with additional panels showing the photographs that are currently relegated to the Supplemental file.

Fig 6. The blue inverted triangles for upper sand ice wedges are very hard to distinguish.

References: Ashastina et al. 2017 Climate of the Past 13: 795-818. Murton et al. 2017 Quat Res 87: 314-330. Opel et al. 2017 Climate of the Past 13: 587-611.

---

## Author Comment (AC2) · 8 Apr 2019

Please see the reply in the attached file. Thomas Opel

Please also note the supplement to this comment:
https://www.clim-past-discuss.net/cp-2018-142/cp-2018-142-AC2-supplement.pdf

---

## Referee Comment (RC3) · Anonymous Referee #3 · 19 May 2019

Manuscript

By: Thomas Opel and co-authors

Journal: Climate of the Past.

The aim of the paper submitted by T. Opel and co-authors to CP is to provide new reconstructions of winter temperatures from Pleistocene relict ice-wedges exposed in the

huge Batagay mega-slump structure in Central Yakutia, completed by a close Holocene outcrop from the Adycha River. This contribution provides new data for the reconstruction and discussion of the palaeoclimate of the Yana highlands on a long time span (< 140 ka or older). The study is based on a short field investigation focusing on the sampling of ice from ice-wedges for isotopic analyse and organic remains from encasing sediments for 14C dating (the oldest part of the sequence being already dated by OSL). The reconstruction of winter palaeotemperatures is achieved using co-isotopic composition ($\delta$D-$\delta$18O) of ice from the ice-wedges preserved in the various Batagay sections and from intra-sedimentary ice.

It is a very interesting contribution based on a methodology that has already been developed and published by the first author in PPP (Opel et al., PPP, 2018). The manuscript is well organised and the topic is fully in agreement with the publication goals of Climate of the Past.

Observations and questions: 1) My main concern is that the author never consider that the climate of the Last glacial was subject to extremely rapid climatic changes between stadial and interstadial conditions that would probably have affected winter temperatures. This point should be discussed. In addition, in the discussion the various episodes of thawing of the ice complex and associated erosional surfaces are only allocated to interglacial conditions (P. 8/L.16) whereas it is likely that they could also occur during MIS 3 during the warming phases characterising the beginning of the stronger interstadials as GI 14 or 12 for example... This has been apparently observed by the authors (P.10/L15): Âń The upward transition of wedges from the upper ice complex to upper sand unit, however, was interrupted episodically by thaw, producing a number of thaw unconformities at different depths Âż.

2) Please explain the difference between Ice complexes and Sand units including intra-sedimentary ice / % of ice?

3) General concern: even if it is not the main topic of the paper, palaeoenvironmental

and palaeoclimatic information derived from vegetal macro-remains should be exposed with more details (only short mentions as in P.9/L.1).

4) According to the various radiocarbon dating results, exposed in part 4.1, no ages corresponding to MIS 2 are found in the upper Ice and upper Sand Complexes presented as dating from MIS3-2 (P.8/L.13) ?. In addition, most of the 14C ages are even older (37 - 38 000 BP or infinite MIS 4?). + P8/L.28-30: "radiocarbon dating of a wood layer (up to 1.5 thick) ABOVE the Lower sand unit and the upper Ice Complex reveal an infinite radiocarbon age ..." According to this statement the upper Ice Complex should not be allocated to MIS3 and even more to MIS 2.

5) As pointed by R2 I also think that the change in isotopic composition of ocean water between glacial periods and the Holocene should be taken in account ...

6) Finally: in the whole paper the water supply, necessary to the development of individual ice-veins forming ice wedges, is supposed to result only from precipitations (snow). What about the proportion of water originating from the melting of markedly older ice-rich permafrost sediments or ice wedges that could be trapped in permafrost cracks and mixed in various proportion with snow melt water during ice wedge forming process?

Figures I have presently great difficulties to connect the very large-scale Figure 3, based on a panoramic photograph, and the schematic but detailed cryo-stratigraphic illustrations of figures 4 & 5. In addition in Figure 3 the author should at least include a vertical scale! and the location of the various studied profiles is not very clear. To resolve this problem we need an additional medium scale Figure (a kind of stratigraphic log) based on a summary of the various formations, including their stratigraphic relations, respective thickness and the position of the various studied sections.

Conclusion This is a new and interesting contribution, well suited for Climate of the Past, and I think that it can be published after a moderate revision taking in account the various questions and observations exposed above.

Please also note the supplement to this comment:
https://www.clim-past-discuss.net/cp-2018-142/cp-2018-142-RC3-supplement.pdf
* * *

---

## Author Response (AR1)

**Reply to editor's comments**

Dear Denis-Didier, we much appreciate your effort and your helpful comments. Our replies are given below in blue.
Please note: All page and line numbers refer to the revised manuscript with track changes after completing the revision based on comments of three referees.

P3, L6 What is the elevation of the site? You never indiacte such parameter to the study sites, please do so
Thank you for this suggestion! We added the elevation for both study sites.

P4, L14 give the coordinates of this site
Thank you for this suggestion! We added coordinates for our Adycha sampling site.

P4, L26 I guess these bags were prepared before to prevent any pollution. Can you precise this point please?
We used freshly opened standard Whirl-Pak bags to store and melt the samples without any additional preparation. We added some information to the text.

P5, L11 How was it extracted?
We picked organic remains from our ice-wedge as well as host sediment samples at the ice-wedge sampling sites. We added this to the manuscript.

P7, L22 add "bs" that you use later
Changed accordingly.

P7, L26 can you be more precise than unidentified plant? Were they leaves, piece of wood, roots? Same thing about the beetle remains.
We added more details to the text (Plants: bract fragments and roots, Beetle remains: complete pieces and fragments of elytron).

P7, L29 idem
We added more details to the text (twigs, roots and florets).

P8, L16 can you check as it seems to be rather Empetrum nigrum? add also the common name "crowberry"
Thank you for correcting our mistake! Changed accordingly.

P8, L19 I would rather first introduce the values and then how they plot on the diagram than the contrary which doesn't seem logical.
Changed accordingly.

P8, L26 are 1.4m above river level a height enough preventing any pollution by the river during flood periods during the melting season of the snow cover?
The sampling site was actively eroding and therefore freshly exposed. Hence, a lateral contamination with river water can be excluded. We cannot exclude flooding of this river bank during the spring snowmelt period but haven't observed any indications on recent flooding on the riverbank surface. We would furthermore expect that the frost crack would have been already filled with snowmelt generated above the crack rather than with potential river flood water. And even if spring flood might contribute to frost crack infill, we assume only minor contamination as the spring flood also originates from winter snow.
We haven't changed the text.

P9, L12 Thanks for adding this comment following reviewer 3's review. I second this review as in the European loess sequences we have studied, thermokarsts have been described corresponding to one of the long interstadials/Dansgaard-Oeschger events identified in Greenland ice-core (see Rousseau et al 2017 Quat. Sci. Rev. fig. 2).

It is indeed one of the major open research questions to elucidate whether these long interstadials/D-O events know from Greenland ice cores and European loess sequences can also be detected with similar characteristics in the remote and highly continental East Siberia. The likely windblown sediments exposed in the Batagay megaslump (in particular the Upper Sand unit) may help to solve this question.

P9, L23 in fact it depends for which purpose you use these dates: dating the ice wedge development or the dating of its infilling?

The timing of frost-crack infilling (i.e. formation of an ice vein within an ice wedge) and ice-wedge development is identical. Please note that we are not speaking about the filling of an ice-wedge pseudomorph by accumulation of new sediments.

P9, L31 but still could it be one of the extremely long younger last climate cycle interstadials than the last interglacial?

This could indeed be, even though it is rather unlikely given the new radiocarbon ages presented in this paper. The unit of wood and plant remains is located about 10 and 5 m below the studied ice wedges B17-IW5 and B17-IW6, respectively. Both, ice wedges and host sediments yielded mainly non-finite ages or ages close to the method limit. Also, the plant macro-fossil record points to interglacial not interstadial conditions. However, to better constrain the age of this unit systematic dating with independent approaches, e.g. luminescence, is needed.
We added a sentence on this to the end of section 5.1.

P11, L34 are you sure that warm=interglacial? interstadial warming in Greenland during the last climate cycle were estimated to be about 12°C in average (see Kindler et al 2013 Clim. Past)

Given the stratigraphic position of this erosional surface and the interpretation of the plant macro-fossil record (see above) we assume rather interglacial than interstadial warming to be the cause for this erosional event. We added an "e.g." to make this clearer. However, additional dating is required for confirmation.

P12, L30 well, this is not clear at all as DO events are associated with high SST linked to variation in the sea ice surface which is not opposed to a changed in the source region of the transported moisture.

We agree and changed the sentence to relate the SST and moisture-source discussion to the Late Holocene (represented by the ice wedge A17-IW3 with much lower $d$ values).

P13, L23 rather write "per mil"
Changed accordingly.

P14, L25 this is Greenland interstadial GI8, one of the long interstadial. See my previous comments.
We agree that there seems to be a temporal coincidence for this warm period. However, in the mentioned East Siberian studies (Wetterich et al., 2014 and Meyer et al., 2002a) the warmer period seems to be longer compared to GI8 (i.e. 48 to 32 kyr, centred around 40ky and regionally varying). No chances were made to the text. Temporally better constrained permafrost deposits from East Siberia are needed to investigate possible relations between the rapid changes in the North Atlantic realm and East Siberia.

P15, L1 Figure 1 doesn't give any indication about the elevation of these mountains. Can you tell?
Both mountain ranges reach elevations of more than 2,000 m. This was added to the text.

P15, L3 is there any modelling experiment available showing this or is this just a working hypothesis?
To our knowledge there is no modelling study available yet, so this is rather a working hypothesis and subject to future work. No changes made to the text.

P15, L16 Is there any modelling experiment supporting this?
To our knowledge there is no modelling study available yet. This is subject to future work. No changes made to the text.

**Reply to referee's comments**
We much appreciate the efforts of all three referees reviewing our manuscript. All reviewers raised some points that will help us to improve our manuscript. Our replies are given below in blue. All page and line numbers refer to the original paper in CPD.

Please note that we have changed the stratigraphic interpretation of ice wedge B17-IW1. Based on new field observations of some of the authors we now attribute ice wedge B17-IW1 to the Lower Ice Complex instead of the Lower Sand as before. The erosional surface indicated in Figure 4 is now interpreted as the stratigraphic boundary between Lower Ice Complex and Lower Sand unit. We have accordingly changed the reply letter and the revised manuscript.

**Anonymous Referee #1**

The authors present a reconstruction of winter precipitation isotope ratios and inferred winter temperatures from relict ice wedges in Pleistocene strata in the Batagay megaslump headwall and late Holocene wedge ice near the Adycha River. These wedges span the last ~140 ka. The authors claim this study fills an important gap in knowledge of paleoclimate of the Yana Highlands. Temporal variations in the isotopic composition of the ice wedges, particularly the wedges that are well-dated and substantial in size (i.e., least prone to post-depositional overprinting), broadly support the conclusion that past stadial and interstadial winters in this region were cooler than today. Further, wedge ice from other areas across Siberia are more enriched than the Yana Highlands, both today and in the past, which suggests this region has always been the most continental area in northern Siberia.
Thank you for this accurate summary!

I read this paper with great interest. Relict ice wedges are an important paleo-meteoric water archive with a tightly constrained seasonality (winter only) and offer some of the oldest known ice in the Northern Hemisphere, extending beyond the temporal limits of the Greenland ice cores. As I see it, ice wedges have an important role to play in advancing our knowledge of Quaternary climate change in non-glaciated Arctic regions where ice cores are not possible.
The authors do a good job explaining their methods and summarising the most important features of this interesting dataset. There are several uncertainties about the integrity of the smaller ice wedge samples, and dating of the pre-MIS 3 wedges, but the authors are up-front about these uncertainties and to a large extent they do not confound the conclusions highlighted in this paper. This paper lays the groundwork for future studies to develop more detailed ice wedge records and paleoclimate inferences from this site. The topic and scope of this work are highly appropriate for Climate of the Past.
I am mostly satisfied with the paper in its current form, but I have a few concerns that I feel should be addressed before it is accepted in final form. Following these revisions I would recommend this paper is accepted.

Thank you for this assessment of our manuscript and for your suggestions and comments we used to improve the manuscript! Please see below for details.

Major comments:
To establish how much more continental the study area was in the past, the authors should consider that the global oceans were more enriched during past cold stages. For example, during MIS2 mean ocean water was ~1.2‰ enriched in 18O. In the discussion, please acknowledge this and provide some discussion – note that standardising for this effect would make some of the relict wedge ice (e.g., B17-IW4) similar in value to the late Holocene wedge ice. I do appreciate there are uncertainties about paleo-source region and possible heterogeneity in the isotope composition of marine source waters. However, some attention to this issue is needed.

Thank you for this suggestion. We agree that this is an important issue when interpreting relict ice isotopes. We, therefore, added this aspect to the discussion in section 5.3. However, we decided not to standardize and recalculate our isotope values. We think that this is not viable because we don't know the exact ages of the studied ice wedges, and so we don't know exactly what sea level was at that time and how isotopically enriched ocean water was exactly.

specific comments
P2, L31 – 'right side' is ambiguous, since it depends on which direction one is oriented.
We replaced "right bank" with "east bank" for clarification.

P3, L13 – 'MIS6 or MIS7'. Does cryostratigraphy provide any indication if this deposit (and the ice wedges) represent a glacial or interglacial period? Ideally say which is more likely.
The cryostratigraphy does not really arbitrate between glacial and interglacial conditions, though the narrow syngenetic wedges and abundance of windblown sand is more consistent with glacial conditions. Palaeoecological data rather indicate the MIS 6 cold stage (Ashastina et al., 2018). We therefore removed the reference to MIS 7 in the entire manuscript.

P4, L1 – you refer to this as the second study site, which is understandable but also confusing since there is a 'Site 2' in Figure 2.
We changed this sentence and omitted "second study site".

P4, L23-25 – if possible, please give a reference for pore ice-wedge ice exchange.
We added two references (Meyer et al, 2002a and Meyer et al., 2010).

P5, L21 – 'outlined below'
Changed accordingly.

P6, L23 – what is meant by redistribution?
We mean relocation of the hare dropping by erosion and deposition processes within the degrading slump which resulted in a contamination. We changed the sentences for clarification.

P11, L5-8 – The co-isotope linearity of this cluster is remarkable. Why not report the slope, intercept and r2 for a line drawn through all data within a cluster, as was done for the individual wedges?
Thank you for this suggestion, we added this information to Table 1 and the text.

P11, L8-9 – arguably, the MIS6 wedge is part of the first cluster. Only one of the datapoints is an outlier.
We assume you refer to ice wedge B17-IW6 from the Upper Ice Complex, not to ice wedge B17-IW1 formerly interpreted as MIS 6. We agree that 3 out of 4 samples of B17-IW6 plot below and close to the GMWL. Therefore, we added B17-IW6 to cluster 1.

P11, L12-15 – this could be explained more. If you are correct, then divergence from the Cluster 1 line could provide valuable information about aridity.

The observed relation between wedge d excess and cryostratigraphic interpretation (high d excess corresponding to wedges formed during drier conditions, lower d excess corresponding to wedges formed during moister conditions) might indeed point towards the divergence from cluster 1 (or GMWL) as indicator of aridity (or drier conditions). This, however, needs thorough evaluation as it might be a site or regional feature and has not been observed in today's coastal ice-wedge sites. We changed the wording in the manuscript for more clarity.

P11, L17 – please add a reference (e.g., Pfahl and Sodeman, 2014, Clim. Past.)

Thank you for this suggestion. We added this reference.

P11, L21 – this is an interesting line of discussion. Can you expand on this point, and comment on how different snowpack evolutions would be expected to influence the isotopic composition of the eventual meltwaters?

Little is known yet about the isotopic changes of the winter precipitation in the snowpack before its incorporation into wedge ice. A few studies suggest an isotopic enrichment in the snowpack and/or the snowmelt prior to filling of frost cracks (see Opel et al., 2018, Permafrost and Periglacial Processes; Grinter et al., 2018, Quaternary Research) due to several processes (e.g. sublimation and depth-hoar development). The effect on $d$ excess hasn't been estimated yet. For intrasedimental ice it is even more unclear as several freeze-thaw cycles have to be taken into account. We, therefore, added only a few words and the references to the text.

P12, L2-5, the point of this sentence is not entirely clear.

We changed this sentence for clarity.

P12, L9, specify that you are talking about d18O. Also, I would advise against specifying a number (e.g., -40) … better to simply say even more depleted values compared to MIS3 wedge ice is expected.

Changed accordingly.

P12, L18-19, please clarify what is meant here.

We deleted this sentence.

P14, L1, clarify that mean dexcess of 6 permille is a Late Holocene value … same for the Yakutia example.

Changed accordingly.

P14, L18-20, this last sentence seems irrelevant to the study … further, it is not clear how this study area provides the ideal conditions are validating/advancing the Cl dating method. Suggest deleting this sentence.

The Batagay megaslump may be a very good site to validate/advance the $^{36}Cl/Cl^-$ dating approach, as it provides both, very old ice and a logical stratigraphy. The latter is missing at other study sites in the Siberian North that exhibit complex spatial stratigraphic patterns. However, as suggested by ref#2 we have moved this aspect to the discussion (section 5.1) and have expanded it slightly.

Figure 7. the #7 datapoint is missing x/y error bars.

We checked this figure again and found that there are error bars for datapoint #7. There are, however, no error bars for the second Batagay datapoint (#16) as these values haven't been reported by Vasil'chuk et al. (2017). We furthermore updated the dataset for several sites and therefore this figure as suggested by ref#2.

**Anonymous Referee #2**

The manuscript presents stratigraphy, geochronology, and ice-wedge stable isotope data from the Batagay megaslump- a remarkable bluff of late-Middle- and and Late-Pleistocene sediments in interior Yakutia exposed by a spectacularly-large thaw slump. They add some new radiocarbon dates to the emerging chronological framework for this site, and provide some new stable O and H isotope data for a small number of ice-wedges ranging in age from MIS 6(?) to the Holocene. Focusing on the broad MIS 3 interval, the authors conclude that winter temperatures during MIS 3 were colder at this site in interior Yakutia, relative to a compilation of purportedly MIS 3 ice wedges from mostly coastal sites.

Thank you for this summary.

In principle I think data and discussion from this kind of proxy archive are a good fit for the scope and audience for Climate of the Past. I also think this site is really a remarkable find, particularly because of the potential for preservation of pre-MIS 5 relict ground ice. The writing and figures are mostly clear. But several factors make me unable to recommend publication: 1 Relatively low number of analyses 2 Poor chronology that inhibits meaningful comparison between sites 3 Speculative nature of the paleoclimate discussion.
I elaborate on these points below, with more specific comments at the end.

Thank you for your assessment, we respond to your more detailed comments below.

Note: Though the title of the manuscript alludes to Middle and Late Pleistocene climate and continentality, my main points of concern are limited to the MIS 3 part of the story because the authors acknowledge that the data from younger and older parts of the sequence are equivocal (p14/line10).

1. The Batagay megaslump headwall exposure is over 1 km long, yet the conclusions re: MIS 3 rest on data collected from only two ice wedges at a single measured section. The discussion and conclusion around Holocene climate is similarly based on analyses of only a single ice wedge. I realize that field work on sites like this is difficult and potentially dangerous, but the chainsaw sampling is rapid and contemporary analytical techniques allow for hundreds of samples to be analyzed in relatively short order. Rather than limited and equivocal results from a reconnaissance visit to the field site, my sense is that this topic deserves "high-resolution systematic sampling and dating", as the authors point out in their conclusion.

We acknowledge that a larger number of studied ice wedges and samples would have been better but unfortunately this could not be realized due to short time in the field and dangerous outcrop conditions, in particular close to the headwall. However, we are convinced that our results and conclusions presented in this study are not equivocal but worth to be published in Climate of the Past (as also stated by referee 1) for the following reasons:
1. The Yana Highlands as most continental region of the Northern Hemisphere are largely understudied in terms of palaeoclimate research.
2. We present stable isotope data of six ice and composite wedges from the Batagay megaslump and one ice wedge from the nearby Adycha River. When compared to previous ice-wedge studies this is not a low number of studied wedges.
3. We present this for the first time in an international peer-reviewed journal and the first data at all for the Lower Ice Complex and a nearby Holocene wedge. Our data for the MIS 3 is confirmed by independent data previously published in Russian by Vasil'chuk et al. (2017) but we provide additional cryostratigraphic context and new radiocarbon age information.
4. Our Batagay MIS3 stable isotope data show unprecedented low $\delta^{18}O$ values and remarkably high $d$ values that are worth to be presented and discussed.
5. Our study is the first one that compares published and yet unpublished ice-wedge stable isotope data for the MIS 3 across Siberia and show that winter temperatures during the MIS 3 have been

lower in the Yana Highlands than in all other regions. A similar pattern is found for the late Holocene and can be deduced for modern winter precipitation.
6. We deduce that the extreme continentality known from modern meteorological observations persisted also during the MIS 3 and the late Holocene.
7. We present new ice-wedge observations and relate them together with the stable isotope composition of ice and composite wedges to the deposition regimes of the different units exposed in the Batagay megaslump.
8. We furthermore present the first pore-ice stable isotope data for the Yana Highlands.

2. Most critically, I question if the available data support a meaningful conclusion re: MIS 3. In other words, what does it actually mean to compare a single probably-MIS3 ice-wedge from one site to another single probably-MIS3 ice-wedge from another site (as is done in Fig 7 and Table S2), since this interval spans ~30,000 years and includes some pretty high-amplitude multi-millennial-scale climate oscillations in high-resolution proxy records?
We already acknowledged in the manuscript that there are some limitations in the dataset used for the spatial comparison:
"Additionally, all except two of the considered wedges or their host sediments have been directly dated by radiocarbon methods to between about 50 and 30 kyr ago. We appreciate that an ice wedge may not contain a full record of this time period and that the climate was not uniform throughout this period; in particular, a warmer period around 40 kyr BP is well known from several palaeoecological proxies for some of the study sites (Wetterich et al., 2014)."

However, we don't know yet if millennial-scale climate oscillations characterized MIS 3 in East Siberia (West Beringia), underlining the need for some high-resolution records covering this period to determine if they did exist. Thus, we are uncertain of their spatial extent and significance in the ice-age Northern Hemisphere. Following Murton et al. (2017), there are indeed hints that the MIS 3 in northern areas of Northeast Siberia was a time of general climate stability. Other records suggest that the interstadial climate was not monolithic, but that the early interstadial was characterized by dry but relatively warm conditions, and cooler and drier climates prevailed later in MIS 3 (e.g. Kienast et al. 2005, Quaternary Research; Sher et al. 2005, Quaternary Science Reviews; Lozhkin et al. 2007, Journal of Paleolimnology; Wetterich et al. 2014, Quaternary Science Reviews). This pattern differs from more southerly sites with evidence for numerous climate fluctuations (Lozhkin&Anderson. 2011, Quaternary Science Reviews).

We added some information on this to section 5.4.1.

To ensure the best possible quality of the dataset
1) we excluded the MIS2 period which shows lower temperatures in Northeast Siberia and limited the time period of interest to 50 to 30 kyr;
2) except for two ice wedges (one of these is also from the Batagay megaslump (Vasil'chuk et al., 2017) we considered only dated ice wedges (radiocarbon dating of host sediments or ice wedges);
3) we reported not only mean values for the respective ice wedges but also standard deviations to account for climate-dependent $\delta^{18}O$ variations captured by the ice wedges;
4) we used only ice wedge datasets that include both $\delta^{18}O$ and $d$ (to be able to detect outliers due to post-depositional processes).

The time interval is also notoriously difficult to date accurately with 14C methods, and many purportedly finite ~35-45 14C ka BP dates in the literature ought to be viewed with a strong dose of skepticism (notably, for example, the purported MIS 3 chronology for Mamontova Gora - an important comparative site Fig 7/Table S2).
To our understanding, 14C dating has made substantial progress over the last years, making 14C dating even in the age range 35-45 k 14C years much more reliable. Calibration is possible back to

50ka cal BP (Reimer et al., 2013, Radiocarbon). All 14C ages used in the manuscript are from the last about 15 years and not from older literature. This is also true for the 14C age from Mamontova Gora (Popp et al., 2006, Permafrost and Periglacial Processes).
Furthermore, we are using 14C to identify the MIS 3 as a whole, not its substages, i.e. MIS 3 rather than Dansgaard-Oeschger events. So, the technique applied is adequate to do this.

Seven out of the 18 sites in the Fig 7/Table S2 ice-wedge compilation are unpublished, so readers can't assess the reliability of these chronologies for themselves.
All stable isotope data used for Figs. 7-9 including previously unpublished data will be made available in PANGAEA after acceptance of the manuscript, except the data from Vasil'chuk et al. (2017).

The authors mention the issue of dating and a long MIS 3 (p12/line26) but do not really address it in a way that justifies the approach. One example of the interpretive difficulties: Novaya Sibir, Belkovsky, and Kotel'ny are all above 74 degN in the New Siberian Islands, yet only Novaya Sibir has relatively depleted isotope composition. Is the between-site difference in isotope composition due to differing age or some sitespecific factor? Either way, the lack of good chronological control inhibits meaningful comparison.
The focus of our study is on the ice wedges from Batagay and we use the other data to illustrate the peculiarity of the Yana Highlands. To distinguish between age- and/or site-specific aspects in the New Siberian Islands is beyond the scope of this paper. By focusing on the broad MIS timescale, the isotope values from Batagay contrast significantly with those from other sites. Despite uncertainties in age the Batagay ice wedges are more depleted than those at any other North Siberian site.

One last point of criticism on the comparison of different sites: why was the compilation/comparison limited to just one "MIS 3" ice-wedge from each site? In the context of this analysis, would it not be more useful to compare the average isotope composition of multiple ice-wedges from a particular stratigraphic interval (e.g. the 10 ice wedges with dD and d18O data attributed to the yedoma ice complex in Opel et al 2017)?
Thank you for this comment. Unfortunately for several sites we rely on only one studied ice wedge. However, we followed the referee's suggestion and reviewed again stable isotope data related to the MIS 3 ice wedges. We added all available data to the following study sites: Lena Delta/Kurungnakh (4), Bykovsky (6), Muostakh (7), Buor Khaya (8), Belkovsky (9), Stolbovoy (10), Bol'shoy Lyakhovsky (12), and Oyogos Yar (13) and updated Figures 7 to 9 and Table S3 accordingly.

3. I acknowledge that quantitative paleoclimate reconstruction from this type of archive is highly uncertain, but the climate implications presented here are vague. Differences in isotope composition between areas are quantified, but then unsupported paleotemperature interpretions are made (e.g. "significantly lower [temperatures]" p14/line7; "extremely low winter temperatures" p11/line23 vs "very low winter temperatures" p11/line26). These distinctions need to be defined.
The paleotemperature interpretation has followed a tentative classification also used in Opel et al. 2017 (Climate of the Past) and is not based on quantitative temperature calibration. We, therefore, decided to omit the use of this paleotemperature interpretation. We have changed the text and Table 4 accordingly.

The authors have not really addressed the issue of paleogeography, nor potential differences in moisture source both through time and for different sites. For example (assuming for a moment that it's possible to meaningfully compare MIS 3 IWs at the Batagay site to those compiled in Fig 7), there's an interesting spatial pattern whereby the Novaya Sibir Island site also has highly depleted IW isotope composition during MIS 3. What are the paleogeographic implications of Late Pleistocene sea level change, with respect to continentality? What are current and modelled MIS 3 moisture sources for that site and the Batagay site? Is there paleoceanographic proxy evidence (e.g. from planktic

forams) for changes in surface water isotopic composition at likely moisture sources? All of these points would likely provide useful context for evaluating the data presented in the manuscript.

During MIS 3 the sea level was lower by 60 to 80 m, the wide Arctic shelves were exposed, leading to an increased continentality of entire Northeast Siberia. However, Novaya Sibir' Island would still have the lowest continentality of all considered sites. To analyze current and modelled MIS3 moisture sources for Batagay and Novaya Sibir' is beyond the scope of the manuscript which is not an MIS 3 stable isotope review but focusing on the cryostratigraphy and new ground ice isotope data from Batagay.

The issue of a changed surface water isotope composition at the moisture sources was also raised by referee #1 (see above). We added some thoughts on this to the discussion.

Other points: This manuscript, which includes many co-authors on earlier papers that document the chronostratigraphic framework for the site, introduces yet another unit-stratigraphic nomenclature for the Batagay megaslump headwall exposure. For example, at least by my interpretation, "upper Ice Complex" (this ms) = Unit III (Ashastina et al. 2017) = Unit 4 (Murton et al. 2017). Given the potential importance of this site, and since all the key players are co-authors on this manuscript, it would be very useful to the community if the authors could reconcile these different frameworks here in this manuscript.

You are right, the different stratigraphies which originate from different studied sections of the outcrop (see Figure 2) can be confusing. But observations of main units are more or less similar.

To relate the stratigraphies used in Ashastina et al. (2017) and Murton et al. (2017) and to establish a reconciled stratigraphy for future studies of the Batagay megaslump we added a new table (Table 1) containing this information.

This group is highly experienced in stable isotope studies of ground ice. Nevertheless, it would be useful to provide some additional methods data. Is the quoted lab precision for dD and d18O 1sigma or 2sigma? What are the summary statistics for the internal quality control secondary standard?

The lab precision at AWI Potsdam isotope lab is given as better than ±0.1‰ for $\delta^{18}O$ and ±0.8‰ for $\delta D$, respectively. For this purpose, we use a three-point calibration for each measurement sequence and one standard for internal quality control. When one of the three standards is outside the ranges given above, the measurement is repeated.

All single measurements have a 1 sigma error, which usually is <0.05‰ for $\delta^{18}O$ and <0.5‰ for $\delta D$. When the single measurement 1 sigma is larger than ±0.1‰ for $\delta^{18}O$ and ±0.8‰ for $\delta D$, the measurement is repeated.

When testing for internal quality control using an independent standard, these tests usually yield a statistical error of <0.05‰ for $\delta^{18}O$ and <0.5‰ for $\delta D$ for long-term measurements (N>20). This is, however, not done for each individual sample series, but tested on a regular base (every few months).

And most importantly (p4/line24), how exactly did the authors decide which samples to exclude from further analyses? The description in the manuscript is very vague (please clarify, with a citation or two, what exchange processes are being invoked between IW and pore ice), and I would strongly prefer that the authors present ALL the ice-wedge stable isotope data first and then justify to readers why some data should be excluded from further consideration.

We first plotted the stable isotope profiles and compared them with our observations of sediment content in the wedge-ice samples. In total, five samples out of the studied seven wedges had to be excluded due to high sediment content and/or distinctly changed isotope values (similar to those of intrasedimental ice of host sediments). For transparency, we decided to present all ice-wedge stable isotope data in the supplement (Table S2) and indicate the excluded samples.

As far as we know these processes haven't been studied in detail. We therefore added two references to the manuscript (Meyer et al. 2002a, 2010) that present similar cases of exchange

processes between wedge ice and host sediments (detectable in stable isotopes and electrical conductivity).

Specific comments:
Referencing: There are points in the introduction and discussion where it would be useful and appropriate to include some citations to relevant work in North America, where there is a long tradition of stable isotope work on ice wedges (e.g. Fraser and Burn; Michel) and the stratigraphic complexities of Middle/Late Pleistocene permafrost exposures (e.g. Péwé, Westgate; Fraser & Burn; Froese, Reyes).
Thank you for this suggestion. We, added some references to the introduction and discussion of chronostratigraphy.

Title: Given the substantial interpretive and chronological uncertainty re: the lower sand and upper unit ice wedge data, I suggest removing "Middle and Late Pleistocene" from the title and replacing it with something more specific
We replaced "Middle and Late Pleistocene" by "Past" to take into account the chronological uncertainty regarding the Lower Ice Complex (was Lower Sand unit before) and consider the Holocene ice-wedge data.

Section 1 in the slump floor: Why is the one sampled wedge from the lower sand collected away from the exposed headwall, as indicated in Fig 3?
Due to the dangerous working conditions at the foot of a 55 m high and partly inclined headwall with thawed material falling down constantly, we decided to sample this wedge in another position close to the headwall.

Can authors reject the possibility that the sampled section is actually younger material displaced into an apparently lower stratigraphic position by slumping?
Yes, we can do this. Several ice wedges were found in the Lower Ice Complex (was lower part of the Lower Sand unit before) in the badlands-like not degraded remains of the slump floor. The material sampled fits well with observations made at the headwall. Also, there was no evidence of slip surfaces observed in or near section 1. The ice-wedge itself was an intact, freshly exposed ice-wedge.

And I'm troubled by the rejection of the 14C age on hare droppings from the ice-wedge. If this was a pristine, freshly exposed ice-wedge, how would younger material be incorporated into the wedge itself? Surely the outermost surface of the wedge ice is removed prior to sampling? And if material "entered into or later froze onto the surface" (p8/line24), doesn't this also imply possible reliability issues with the isotope data from that wedge?
Given the IRSL age reported in Ashastina et al. (2017) and the erosional surface in Section 1 we can exclude a younger age than MIS 6 for this sample. Therefore, the young age has to be caused by contamination. We see only one possible way: contamination of a non-finite age of the hare dropping by modern carbon during the handling of the ice-wedge sample (i.e. modern tiny and/or dissolved carbon from e.g. mud attached to the hare dropping in the sample bag.
As the ice wedge itself is melting constantly it is somehow "cleaning itself" and the only possible contamination would be with own meltwater.

p8/line18: Do you mean ". . ..different stratigraphic contexts"? This would make more sense. Also, the next few sentences of this paragraph are pretty vague and not particularly useful. I think it's pretty obvious now that adequate dating of these sediments is going to be a major challenge. There's some mention of alternative approaches in the conclusion section, which really should be moved into the discussion and properly addressed.
We added some discussion on dating old permafrost and ice wedges to section 5.1.

wood layer and the thaw unconformity (p. 8/9 transition): I assume you mean ". . .situated above the lower sand unit and BELOW the upper Ice Complex. . ..." on p8/line29, since you reasonably attribute the wood layer to the last interglacial?
You are right, changed accordingly.

14C dating (Table 2 and p5): I appreciate the details on pre-treatement and analysis. Please clarify if smaller blanks were measured for background correction of the many samples with low mass of organic C (Table 2).
Size-matched standards and blanks were analysed along with the samples and were used for normalization and blank correction. We added this information to the text.

Fig. 5: The changing vertical scale is confusing. Since the ice-wedge morphology is important in this context, I recommend a stratigraphic diagram to scale, with additional panels showing the photographs that are currently relegated to the Supplemental file.
We considered this suggestion. As a stratigraphic diagram to scale would look spartan, we decided to keep the original diagram but to add indications for the broken vertical scale. We feel, that the ice-wedge photographs are well placed in the supplement.

Fig 6. The blue inverted triangles for upper sand ice wedges are very hard to distinguish.
We changed the colors to improve visible distinguishability.

References: Ashastina et al. 2017 Climate of the Past 13: 795-818. Murton et al. 2017 Quat Res 87: 314-330. Opel et al. 2017 Climate of the Past 13: 587-611.

**Anonymous Referee #3**

The aim of the paper submitted by T. Opel and co-authors to CP is to provide new reconstructions of winter temperatures from Pleistocene relict ice-wedges exposed in the huge Batagay mega-slump structure in Central Yakutia, completed by a close Holocene outcrop from the Adycha River. This contribution provides new data for the reconstruction and discussion of the palaeoclimate of the Yana highlands on a long time span (< 140 ka or older). The study is based on a short field investigation focusing on the sampling of ice from ice-wedges for isotopic analyse and organic remains from encasing sediments for 14C dating (the oldest part of the sequence being already dated by OSL). The reconstruction of winter palaeotemperatures is achieved using co-isotopic composition (δD-δ18O) of ice from the ice-wedges preserved in the various Batagay sections and from intra-sedimentary ice.
It is a very interesting contribution based on a methodology that has already been developed and published by the first author in PPP (Opel et al., PPP, 2018).
The manuscript is well organised and the topic is fully in agreement with the publication goals of Climate of the Past.
Thank you for this summary!

Observations and questions:
1) My main concern is that the author never consider that the climate of the Last glacial was subject to extremely rapid climatic changes between stadial and interstadial conditions that would probably have affected winter temperatures.
This point should be discussed.

As pointed out already in response to Referee #2 it is not clear yet whether the rapid climate changes known from the North Atlantic region (e.g. the Greenland ice cores) and Europe have also affected East Siberia (West Beringia) due to the lack of high-resolution records covering this period. Thus, we are uncertain of their spatial extent and significance in the ice-age Northern Hemisphere. Following

Murton et al. (2017), there are indeed hints that the MIS 3 in northern areas of Northeast Siberia was a time of general climate stability. Other records suggest that the interstadial climate was not monolithic, but that the early interstadial was characterized by dry but relatively warm conditions, and cooler and drier climates prevailed later in MIS 3 (e.g. Kienast et al. 2005, Quaternary Research; Sher et al. 2005, Quaternary Science Reviews; Lozhkin et al. 2007, Journal of Paleolimnology; Wetterich et al. 2014, Quaternary Science Reviews). This pattern differs from more southerly sites with evidence for numerous climate fluctuations (Lozhkin&Anderson. 2011, Quaternary Science Reviews).
We added some information on this to section 5.4.1.

In addition, in the discussion the various episodes of thawing of the ice complex and associated erosional surfaces are only allocated to interglacial conditions (P. 8/L.16) whereas it is likely that they could also occur during MIS 3 during the warming phases characterising the beginning of the stronger interstadials as GI 14 or 12 for example…
This has been apparently observed by the authors (P.10/L15): « The upward transition of wedges from the upper ice complex to upper sand unit, however, was interrupted episodically by thaw, producing a number of thaw unconformities at different depths ».
Thank you, we added the exceptionally warm interstadials as another possible cause for the observed thaw unconformities.

2) Please explain the difference between Ice complexes and Sand units including intrasedimentary ice / % of ice?
Unfortunately, there are no data on gravimetric ice content available yet. This will be a subject of future studies. The major difference is the amount of wedge ice: huge and wide ice wedges in the Ice Complexes and narrow composite wedges in the sand units.

3) General concern: even if it is not the main topic of the paper, palaeoenvironmental and palaeoclimatic information derived from vegetal macro-remains should be exposed with more details (only short mentions as in P.9/L.1).
We added some information here as this unit of wood and plant remains does not contain ice wedges and is therefore not considered in later parts of the manuscript. However, the main palaeoecological results from Ashastina et al. (2018, Quaternary Science Reviews) are summarized per unit and presented in Table 4 (was Table 3 before) and also mentioned in section 5.3. We prefer not to include more palaeoecological information to the manuscript which focuses on ground ice isotopes and cryostratigraphy.

4) According to the various radiocarbon dating results, exposed in part 4.1, no ages corresponding to MIS 2 are found in the upper Ice and upper Sand Complexes presented as dating from MIS3-2 (P.8/L.13) ?.
You are right, most of the new radiocarbon ages presented in this study belong to MIS 3 (except for B17-IW5-02 which is placed at the boundary of MIS 3 and MIS 2). However, in the earlier study by Ashastina et al. (2017, Climate of the Past) radiocarbon ages for their Section C (equivalent to our Section 2) range from > 51 to 12.66 kyr BP. This supports a deposition of the Upper Ice Complex and the Upper Sand during MIS 3 and MIS 2.

In addition, most of the 14C ages are even older (37 - 38 000 BP or infinite MIS 4?).
+ P8/L.28-30: "radiocarbon dating of a wood layer (up to 1.5 thick) ABOVE the Lower sand unit and the upper Ice Complex reveal an infinite radiocarbon age …"
According to this statement the upper Ice Complex should not be allocated to MIS3 and even more to MIS 2.
There was a mistake. It should read as follows "…above the Lower Sand unit and BELOW the Upper Ice Complex." Hence, we stick to our chronostratigraphic interpretation of the Upper Ice Complex

and Upper Sand being formed during MIS 3 and MIS2. However, we don't really have a firm maximum age for it the Upper Ice Complex at present. A formation beginning in MIS 4 may be possible but has to be the subject of future studies.

5) As pointed by R2 I also think that the change in isotopic composition of ocean water between glacial periods and the Holocene should be taken in account …
The issue of a changed surface water isotope composition at the moisture sources was also raised by referee #1 (see above). We added some thoughts on this to the discussion.

6) Finally: in the whole paper the water supply, necessary to the development of individual ice veins forming ice wedges, is supposed to result only from precipitations (snow). What about the proportion of water originating from the melting of markedly older ice-rich permafrost sediments or ice wedges that could be trapped in permafrost cracks and mixed in various proportion with snow melt water during ice wedge forming process?
The substantial contribution of meltwater of relict ground ice seems to be a very unlikely explanation and has to our knowledge not been described in ice-wedge literature. When ice-wedge cracks are open in winter and spring the active layer is either frozen or just starting to melt, so it seems difficult to envisage how meltwater from old ice-rich permafrost could infill cracks. If the melting occurred in mid to late summer when the active layer was thawed more deeply, then it would be easier to envisage melt of older ground ice, e.g. by underground thermal erosion. But by that time of year the cracks would probably have infilled or closed by thermal expansion.

Figures
I have presently great difficulties to connect the very large-scale Figure 3, based on a panoramic photograph, and the schematic but detailed cryo-stratigraphic illustrations of figures 4 & 5.
In addition in Figure 3 the author should at least include a vertical scale! and the location of the various studied profiles is not very clear.
To resolve this problem we need an additional medium scale Figure (a kind of stratigraphic log) based on a summary of the various formations, including their stratigraphic relations, respective thickness and the position of the various studied sections.
Thank you for this suggestion. We added a stratigraphic scheme to Figure 3 that shows the different units their respective thicknesses and indicated the positions of the sections.

Conclusion
This is a new and interesting contribution, well suited for Climate of the Past, and I think that it can be published after a moderate revision taking in account the various questions and observations exposed above.
Thank you for this assessment!

[revised manuscript text omitted]

**MIS 3 ice wedges**

1 Western Laptev Sea, Mamontov Klyk
2 Lena Delta, Nagym
3 Lena Delta, Khardang
4 Lena Delta, Kurungnakh
5 Lena Delta, Sobo Sise
6 Central Laptev Sea, Bykovsky
7 Central Laptev Sea, Muostakh
8 Central Laptev Sea, Buor Khaya
9 New Siberian Islands, Belkovsky
10 New Siberian Islands, Stolbovoy
11 New Siberian Islands, Kotel'ny
12 New Siberian Islands, Bol'shoy Lyakhovsky
13 Dmitry Laptev Strait, Oyogos Yar
14 New Siberian Islands, Novaya Sibir
15 Lower Kolyma, Duvanny Yar
16 Yana Highlands, Batagay
17 Central Yakutia, Tanda
18 Central Yakutia, Mamontova Gora

**Figure 7.** Comparison of stable isotope data from Siberian ice wedges attributed to the  MIS 3 interstadial (about 50 to 30 kyr), comprising data from the  western Laptev Sea (Magens, 2005), the Lena River Delta (Schirrmeister et al., 2003; Schirrmeister et al., 2011a; Wetterich et al., 2008; Opel, unpublished), the  central Laptev Sea (Meyer et al., 2002a; Meyer/Opel, unpublished; Schirrmeister et al., 2017), the New Siberian Islands and the Dmitry Laptev Strait (Schirrmeister, unpublished; Wetterich et al., 2014; Opel et al., 2017b), the Kolyma Lowland (Strauss, 2010), the Yana Highlands (this study; Vasil'chuk et al., 2017), and  central Yakutia (Schirrmeister, unpublished; Popp et al., 2006). The new data from Batagay are marked in red. Further information is given in Table S3.

[Figure]

**Figure 8**. Map of mean δ¹⁸O data from Siberian ice wedges attributed to the  MIS 3 interstadial (about 50 to 30 kyr), Site IDs are given in **Figure 7** and further details in **Table S3**.

[Figure]

**Figure 9.** Map of mean *d* data from Siberian ice wedges attributed to the  MIS 3 interstadial (about 50 to 30 kyr), Site IDs are given in **Figure 7** and further details in **Table S3**.